# Rethinking conditional GAN training: An approach using geometrically structured latent manifolds

**Sameera Ramasinghe**
The Australian National University, Data61-CSIRO
sameera.ramasinghe@anu.edu.au

**Moshiur Farazi**
Data61-CSIRO

**Salman Khan**
Mohamed Bin Zayed University of AI

**Nick Barnes**
The Australian National University

**Stephen Gould**
The Australian National University

## Abstract

Conditional GANs (cGAN), in their rudimentary form, suffer from critical drawbacks such as the lack of diversity in generated outputs and distortion between the latent and output manifolds. Although efforts have been made to improve results, they can suffer from unpleasant side-effects such as the topology mismatch between latent and output spaces. In contrast, we tackle this problem from a geometrical perspective and propose a novel training mechanism that increases both the diversity and the visual quality of a vanilla cGAN, by systematically encouraging a bi-lipschitz mapping between the latent and the output manifolds. We validate the efficacy of our solution on a baseline cGAN (i.e., Pix2Pix) which lacks diversity, and show that by only modifying its training mechanism (i.e., with our proposed Pix2Pix-Geo), one can achieve more diverse and realistic outputs on a broad set of image-to-image translation tasks. Code available at: https://github.com/samgregoost/Rethinking-CGANs

## 1 Introduction

Generative adversarial networks (GAN) are a family of deep generative models that learn to model data distribution $\mathcal{Y}$ from random latent inputs $z \sim \mathcal{Z}$ using a stochastic generator function $G : \mathcal{Z} \to \mathcal{Y}$ [1]. A seemingly natural extension from unconditional GANs to conditional GANs (cGAN) can be achieved via conditioning both the discriminator and the generator on a conditioning signal $x \sim \mathcal{X}$. However, such a straightforward extension can cause the models to disregard $x$ [2, 3, 4, 5]. To overcome this unsought behavior, a reconstruction loss is typically added to the objective function to penalise the model when it deviates from $x$. This approach has been widely adapted for diverse tasks including image-to-image translation [6, 2], style transfer [7, 8] and inpainting [9, 3, 10], and super-resolution [11, 12, 13, 14]. However, in spite of the wide usage, naively coupling the reconstruction and the adversarial objectives entails undesirable outcomes as discussed next.

Many conditional generation tasks are ill-posed (many possible solutions exist for a given input), and an ideal generator should be able to capture *one-to-many* mappings between the input and output domains. Note that the stochasticity of $G$ typically depends on two factors, *first* the randomness of $z$ and *second* the dropout [15]. However, empirical evidence suggests the composition of reconstruction and adversarial losses leads to a limited diversity, despite the random seed $z$. In fact, many prior works have reported that the generator often tends to ignore $z$, and learns a deterministic mapping

35th Conference on Neural Information Processing Systems (NeurIPS 2021).

from $\mathcal{X}$ to $\mathcal{Y}$, leaving dropout as the only source of stochasticity [2, 4, 3, 5]. Additionally, [16] and [17] demonstrated that from a geometrical perspective, latent spaces of generative models (e.g., cGANs) tend to give a distorted view of the generated distribution, thus, the Euclidean paths on the latent manifold do not correspond to the geodesics (shortest paths) on the output manifold. This hinders many possibilities such as clustering in the latent space, better interpolations, higher interpretability and ability to manipulate the outputs. We show that the foregoing problems can be direct consequences of the conventional training approach. Moreover, the naive coupling of regression loss and the adversarial loss can also hamper the visual quality of the generated samples due to contradictory goals of the two objective functions (see Sec. 2.1).

The aforementioned drawbacks have led multi-modal conditional generation approaches to opt for improved objective functions [18, 19], and even complex architectures compared to vanilla cGANs [20, 5, 4]. However, in Sec. 2, we show that while the existing solutions may improve the diversity and address the loss mismatch, they can also aggravate the topology mismatch and distortion between the latent and output manifolds. In contrast, we argue that these issues are not a consequence of the model capacities of vanilla cGANs [2, 3, 21, 6], rather a result of sub-optimal training procedures that are insensitive to their underlying geometry. As a remedy, we show that the foregoing problems can be addressed by systematically encouraging a structured bijective and a continuous mapping, i.e., a homeomorphism, between the latent and the generated manifolds. Furthermore, the structure of the latent space can be enhanced by enforcing bi-lipschitz conditions between the manifolds. To this end, we introduce a novel training procedure and an optimization objective to encourage the generator and the latent space to preserve a bi-lipschitz mapping, while matching the Euclidean paths in the latent space to geodesics on the output manifold.

We choose Pix2Pix [2], a vanilla cGAN, and modify its training procedure to demonstrate that the proposed mapping improves the realism of the outputs by removing the loss mismatch, enhances the structure of the latent space, and considerably improves the output diversity. As the formulation of our conditional generation approach is generic, we are able to evaluate the modified Pix2Pix model, dubbed *Pix2Pix-Geo*, on a diverse set of popular image-to-image translation tasks. We show that with the modified training approach, our Pix2Pix-Geo significantly improves the prediction diversity of the cGAN compared to the traditional baseline procedure and achieves comparable or better results than the more sophisticated *state-of-the-art* models. Most importantly, our modifications are purely aimed at the optimization procedure, which demands no architectural modifications to vanilla cGANs.

## 2   Motivation

In conditional generative modeling, the ground truth (output) data distribution $\mathcal{Y} \subseteq \mathbb{R}^M$ is conditioned on an input distribution $\mathcal{X} \subseteq \mathbb{R}^d$. Consider the data distribution $\mathcal{Y}_{|x_p} \subset \mathcal{Y}$ conditioned on $x_p \in \mathcal{X}$. Then, the following adversarial objective function is used to optimize the generator $G$ by playing a min-max game against a discriminator $D$, thereby approximating the distribution $\mathcal{Y}_{|x_p}$,

$$L_{adv} = \min_G \max_D \mathop{\mathbb{E}}_{y \sim \mathcal{Y}}[\Phi(D(x_p, y))] + \mathop{\mathbb{E}}_{z \sim \zeta}[\Phi(1 - D(x_p, G(x_p, z)))], \tag{1}$$

where $\Phi$ is a suitably chosen monotone function, $y \sim \mathcal{Y}$ and $z \in \mathbb{R}^k$ is a latent vector sampled from a prior distribution $\zeta$. It has been widely observed that using the above objective function in isolation, pushes the models to generate samples that are not strongly conditioned on the input signal $x_p$ [2, 18, 4, 20]. Hence, the conventional cGAN loss couples a reconstruction loss $L_r$ (typically $\ell_1$ or $\ell_2$) with Eq. 1. However, as alluded in Sec. 1, this entails several drawbacks: **a)** contradictory goals of the loss components, **b)** conditional mode collapse, and **c)** insensitivity to the underlying manifold geometry. Below, we explore these issues in detail and contrast our method against several recent attempts towards their resolution. From this point onwards, our analysis is focused on the conditional setting and we do not explicitly denote the conditioning signal $x$ in our notations, to avoid clutter.

### 2.1   Mismatch b/w adversarial & reconstruction losses

Given the generated distribution $p_g$ and the ground truth distribution $p_d$, the optimal generator $G^*$ for the adversarial loss can be formulated as,

$$G^* = \operatorname*{argmin}_G \Big( \mathbf{JSD}\big[p_g(\bar{y}) \| p_d(y)\big] \Big), \tag{2}$$

where **JSD** is the Jensen–Shannon divergence, $y$ is the ground-truth and $\bar{y} = G(z)$ is the output. Let us also consider the expected $\ell_1$ loss $L_r = \mathbb{E}_{y,z}|y - \bar{y}|$. App. F shows that $L_r$ is minimized when,

$$\int_{-\infty}^{\bar{y}} p_d(y)dy = \int_{\bar{y}}^{\infty} p_d(y)dy. \tag{3}$$

This shows the probability mass to the left of $\bar{y}$ is equal to the probability mass of right of $\bar{y}$, i.e, $\bar{y}$ is the median of $y$. Therefore, the optimal generator obtained from minimizing $L_r$ does not equal to $G^*$, except for the rare case where $p_d(y)$ is unimodal with a sharp peak. With a similar approach, it can be shown that $\ell_2$ concentrates $p_g$ near the average of the ground truth distribution. Hence, these contradictory goals of $L_r$ and $L_{adv}$ force the model to reach a compromise, thereby settling in a sub-optimal position in the parameter space. On the contrary, this mismatch can be removed by our proposed training approach by encouraging a homeomorphism between the latent and output spaces (App. F). This argument is empirically backed by our experiments, as we show that the realism of the outputs of the Pix2Pix [2] model can be significantly improved using the proposed method. Both Bicycle-GAN [20] and MR-GAN [4] remove this loss mismatch using a bijective mapping and by matching the moments of the generated and target distributions, respectively. However, their training procedures can disrupt the structure of the latent space (see Sec. 2.3).

## 2.2 Conditional mode collapse

(Conditional) mode collapse is a commonly observed phenomenon in cGANs [2, 4, 3, 5]. In this section, we discuss how the traditional training procedure may cause mode collapse and show that the existing solutions tend to distort the structure of the latent manifold.

**Definition 1** [18]. A mode $\mathcal{H}$ is a subset of $\mathcal{Y}$ s.t. $\max_{y \in \mathcal{H}} \|y - y^*\| < \alpha$ for an output $y^*$ and $\alpha > 0$. Then, at the training phase, $z_1$ is attracted to $\mathcal{H}$ by $\epsilon$ from an optimization step if $\|y^* - G_{\theta(t+1)}(z_1)\| + \epsilon < \|y^* - G_{\theta(t)}(z_1)\|$, where $\theta(t)$ are the parameters of $G$ at time $t$.

**Proposition 1** [18]. Suppose $z_1$ is attracted to $\mathcal{H}$ by $\epsilon$. Then, there exists a neighbourhood $\mathcal{N}(z_1)$ of $z_1$, such that $z$ is attracted to $\mathcal{H}$ by $\epsilon/2, \forall z \in \mathcal{N}(z_1)$. Furthermore, the radius of $\mathcal{N}(z_1)$ is bounded by an open ball of radius $r$ where the radius is defined as,

$$r = \epsilon \Big( 4 \inf_z \big\{ \max(\tau(t), \tau(t+1)) \big\} \Big)^{-1}, \text{ where } \tau(t) = \frac{\|G_{\theta(t)}(z_1) - G_{\theta(t)}(z)\|}{\|z_1 - z\|}. \tag{4}$$

Proposition 1 yields that by maximizing $\tau(t)$ at each optimization step, one can avoid mode collapse. Noticeably, the traditional training approach does not impose such a constraint. Thus, $\|z_1 - z\|$ can be arbitrary large for a small change in the output and the model is prone to mode collapse. As a result, DSGAN [18], MS-GAN [19] and MR-GAN [4] (implicitly) aim to maximize $\tau$. Although maximizing $\tau$ improves the diversity, it also causes an undesirable side-effect, as discussed next.

## 2.3 Loss of structure b/w output & latent manifolds

A *sufficiently smooth* generative model $G(z)$ can be considered as a surface model [22]. This has enabled analyzing *latent variable generative models* using Riemannian geometry [23, 24, 16, 25]. Here, we utilize the same perspective: a generator can be considered as a function that maps low dimensional latent codes $z \in \mathcal{M}_z \subseteq \mathbb{R}^k$ to a data manifold $\mathcal{M}_y$ in a higher dimensional space $\mathbb{R}^M$ where $\mathcal{M}_z$ and $\mathcal{M}_y$ are Riemannian manifolds, i.e., $z$ encodes the intrinsic coordinates of $\mathcal{M}_y$. Note that increasing $\tau$ in an unconstrained setting does not impose any structure in the latent space. That is, since the range of $\|G(z_1) - G(z)\|$ is arbitrary in different neighbourhoods, stark discontinuities in the output space can occur, as we move along $\mathcal{M}_z$. Further note that Bicycle-GAN also does not impose such continuity on the mapping. Thus, the distance between two latent codes on $\mathcal{M}_z$ may not yield useful information such as the similarity of outputs. This is a significant disadvantage, as we expect the latent space to encode such details. Interestingly, if we can induce a continuous and a bijective mapping, i.e., a homeomorphism between $\mathcal{M}_y$ and $\mathcal{M}_z$, while maximizing $\tau$, the structure of the latent space can be preserved to an extent.

However, a homeomorphism does not reduce the distortion of $\mathcal{M}_y$ with respect to $\mathcal{M}_z$. In other words, although the arc length between $z_1$ and $z$ is smoothly and monotonically increasing with the arc length between $G(z_1)$ and $G(z)$ under a homeomorphism, it is not bounded. This can cause heavy distortions between the manifolds. More formally, maximizing $\tau$ encourages maximizing

the components of the Jacobian $\mathbf{J}^{d \times k} = \frac{\partial G}{\partial z}$ at small intervals. If $G$ is sufficiently smooth, the Riemannian metric $\mathbf{M} = \mathbf{J}^T \mathbf{J}$ can be obtained, which is a positive definite matrix that varies smoothly on the latent space. Further, by the Hadamard inequality,

$$\det(\mathbf{M}) \leq \prod_{i=0}^{k} \|\mathbf{J}_i\|^2, \tag{5}$$

where $\mathbf{J}_i$ are the columns of $\mathbf{J}$. This leads to an interesting observation. In fact, $\det(\mathbf{M})$ can be seen as a measure of distortion of the output manifold with respect to the latent manifold. Therefore, although maximizing $\tau$ acts as a remedy for mode collapse, even under a homeomorphism, it can increase the distortion between $\mathcal{M}_z$ and $\mathcal{M}_y$.

In conditional generation tasks, it is useful to reduce the distortion between the manifolds. Ideally, we would like to match the Euclidean paths on $\mathcal{M}_z$ to geodesics on $\mathcal{M}_y$, as it entails many advantages (see Sec. 1). Consider a small distance $\Delta z$ on $\mathcal{M}_z$. Then, the corresponding distance in $\mathcal{M}_y$ can be obtained using Taylor expansion as,

$$G(\Delta z) = \mathbf{J}\Delta z + \Theta(\|\Delta z\|) \approx \mathbf{J}\Delta z, \tag{6}$$

where $\Theta(\|\Delta z\|)$ is a function which approaches zero more rapidly than $\Delta z$. It is evident from Eq. 6 that the corresponding distance on $\mathcal{M}_y$ for $\Delta z$ is governed by $\mathbf{J}$. Ideally, we want to constrain $\mathbf{J}$ in such a way that small Euclidean distances $\Delta z$ encourage the output to move along geodesics in $\mathcal{M}_y$. However, since random sampling does not impose such a constraint on $\mathbf{J}$, the traditional training approach and the existing solutions fail at this. Interestingly, it is easy to deduce that geodesics avoid paths with high distortions [26]. Recall that minimizing $\tau$ along optimization curves reduces the distortion of $\mathcal{M}_y$, thus, encourages $\Delta z$ to match geodesics on $\mathcal{M}_y$. However, minimizing $\tau$ can also lead to mode collapse as discussed in Sec. 2.2.

Although the above analysis yields seemingly contradictory goals, one can achieve both by establishing a bi-lipschitz mapping between $\mathcal{M}_y$ and $\mathcal{M}_z$, as it provides both an upper and a lower-bound for $\tau$. Such a mapping between $\mathcal{M}_z$ and $\mathcal{M}_y$ provides a soft bound for $\det(\mathbf{M})$, and prevents mode collapse while preserving structure of the latent manifold.

**Remark 1:** *An ideal generator function should be homeomorphic to its latent space. The structure of the latent space can be further improved by inducing a bi-lipschitz mapping between the latent space and generator function output.*[1]

Based on the above Remark, we propose a training approach that encourages a structured bi-lipschitz mapping between the latent and the generated manifolds and show that in contrast to the existing methods, the proposed method is able to address all three issues mentioned above.

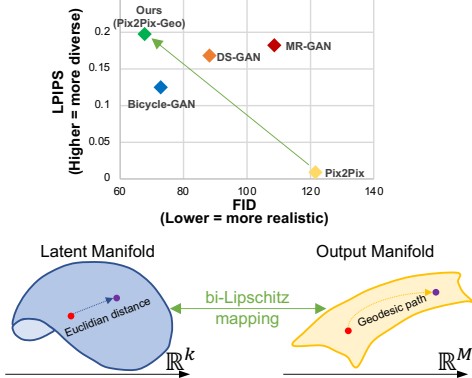

Figure 1: **Approach Overview.** Our training procedure encourages a bi-lipschitz mapping between the latent and generated output manifolds, while mapping the Euclidean shortest paths in the latent manifold to geodesics on the generated output manifold, which allows better diversity and structure. We gain a considerable improvement in both visual quality and the image diversity over our baseline Pix2Pix [2], using the same network architecture (*landmark $\rightarrow$ faces* image-to-image translation task).

## 3   Methodology

Our approach is based on three goals. **R1**) A bi-lipschitz mapping must exist between $\mathcal{M}_z$ and $\mathcal{M}_y$,

$$\frac{1}{C}d_{\mathcal{M}_z}(z^p, z^q) \leq d_{\mathcal{M}_y}(\phi^{-1}(z^p), \phi^{-1}(z^q)) \leq C d_{\mathcal{M}_z}(z^p, z^q), \tag{7}$$

where $d_.(\cdot)$ is the geodesic distance in the denoted manifold, $z^p$ and $z^q$ are two latent codes, and $C$ is a constant. Further, $\phi : \mathcal{M}_y \rightarrow \mathcal{M}_z$ is a continuous global chart map with its inverse $\phi^{-1}$. **R2**) Euclidean distances in $\mathcal{M}_z$ should map to geodesics in $\mathcal{M}_y$ for better structure. **R3**) The geodesic distance between two arbitrary points on $\mathcal{M}_y$ should correspond to a meaningful metric, i.e., pixel distance (note the loss mismatch is implicitly resolved by **R1**). Next, we explain our training procedure.

---

[1]Note that every bi-lipschitz mapping is a homeomorphism.

## 3.1 Geodesics and global bi-lipschitz mapping

Here, we discuss the proposed training procedure. Consider a map $\gamma_{\mathcal{M}_z} : I \to \mathcal{M}_z$, that parameterizes a curve on $\mathcal{M}_z$ using $t \in I \subset \mathbb{R}$. Then, there also exists a map $(G \circ \gamma_{\mathcal{M}_z}) \equiv \gamma_{\mathcal{M}_y} : I \to \mathcal{M}_y$. If $\gamma_{\mathcal{M}_y}$ is a geodesic, this mapping can be uniquely determined by a $p \in \mathcal{M}_y$ and an initial velocity $V \in T_p \mathcal{M}_y$, where $T_p \mathcal{M}_y$ is the tangent space of $\mathcal{M}_y$ at $p$ (see App. C)[2]. This is a useful result, as we can obtain a unique point $p' \in \mathcal{M}_y$ only by defining an initial velocity and following $\gamma_{\mathcal{M}_y}$ for a time $T$ (note we do not consider the unlikely scenario where two geodesics may overlap at $t = T$).

To find the geodesic between two points on a Riemannian manifold, $\gamma_{\mathcal{M}_z}$ is usually constrained as,

$$\ddot{\gamma}_{\mathcal{M}_z} = -\frac{1}{2}\mathbf{M}^{-1}\left[2(\mathbf{I}_k \otimes \dot{\gamma}_{\mathcal{M}_z}^T)\frac{\partial \mathrm{vec}(\mathbf{M})}{\partial \gamma_{\mathcal{M}_z}}\dot{\gamma}_{\mathcal{M}_z} - \left[\frac{\partial \mathrm{vec}(\mathbf{M})}{\partial \gamma_{\mathcal{M}_z}}\right]^T (\dot{\gamma}_{\mathcal{M}_z} \otimes \dot{\gamma}_{\mathcal{M}_z})\right],$$

where $\mathbf{M}^{k \times k} = \mathbf{J}_{\phi^{-1}}^T \mathbf{J}_{\phi^{-1}}$ is the metric tensor, $\mathbf{J}_{\phi^{-1}}$ is the Jacobian, $\otimes$ is the outer product, dot operator is the first-order gradient and the double dot operator is the second-order gradient [17]. This approach is expensive, as it requires calculating the Jacobians in each iteration and moreover, causes unstable gradients. In practice, an exact solution is not needed, hence, we adapt an alternate procedure to encourage $\gamma_{\mathcal{M}_y}$ to be a geodesic, and use Eq. 8 only for evaluation purposes in Sec. 4. Since geodesics are locally length minimizing paths on a manifold, we encourage the model to minimize the curve length $L(\gamma_{\mathcal{M}_y}(t))$ on $\mathcal{M}_y$ in the range $t = [0, T]$. $L(\gamma_{\mathcal{M}_y}(t))$ is measured as:

$$L(\gamma_{\mathcal{M}_y}(t)) = \int_0^1 \left\|\frac{\partial G \circ \gamma_{\mathcal{M}_z}(t)}{\partial t}\right\| dt = \int_0^1 \left\|\frac{\partial G \circ \gamma_{\mathcal{M}_z}(t)}{\partial \gamma_{\mathcal{M}_z}(t)}\frac{\partial \gamma_{\mathcal{M}_z}(t)}{\partial t}\right\| dt. \tag{8}$$

Eq. 8 can be expressed using the Jacobian $\mathbf{J}_{\phi^{-1}}$ as,

$$= \int_0^1 \left\|\mathbf{J}_{\phi^{-1}}\frac{\partial \gamma_{\mathcal{M}_z}(t)}{\partial t}\right\| dt = \int_0^1 \sqrt{\left[\mathbf{J}_{\phi^{-1}}\frac{\partial \gamma_{\mathcal{M}_z}(t)}{\partial t}\right]^T \mathbf{J}_{\phi^{-1}}\frac{\partial \gamma_{\mathcal{M}_z}(t)}{\partial t}} dt.$$

Since $\mathbf{M}^{k \times k} = \mathbf{J}_{\phi^{-1}}^T \mathbf{J}_{\phi^{-1}}$,

$$= \int_0^1 \sqrt{\left[\frac{\partial \gamma_{\mathcal{M}_z}(t)}{\partial t}\right]^T \mathbf{M}\frac{\partial \gamma_{\mathcal{M}_z}(t)}{\partial t}} dt.$$

Considering small $\Delta t = \frac{T}{N}$,

$$\approx \sum_{i=0}^N \sqrt{\left[\frac{\partial \gamma_{\mathcal{M}_z}(t)}{\partial t}\right]^T \mathbf{M}\frac{\partial \gamma_{\mathcal{M}_z}(t)}{\partial t}}\Delta t = \sum_{i=0}^{N-1} \sqrt{\dot{z}_i^T \mathbf{M}\dot{z}_i}\Delta t. \tag{9}$$

Further, $\left\|G(\Delta z)\right\| = \Delta z^T \mathbf{M}\Delta z > 0, \forall \Delta z > 0$, i.e., $\mathbf{M}$ is positive definite (since $\frac{dG}{dz} \neq 0$, which is discussed next). By Hadamard inequality (Eq. 5), it can be seen that we can minimize the $\frac{\partial G}{\partial z}$, in order for $L(\gamma_{\mathcal{M}_y}(t))$ to be minimized. But on the other hand, we also need $\gamma_{\mathcal{M}_y}(T) = y$. Therefore, we minimize $\frac{\partial G}{\partial z}$ at small intervals along the curve by updating the generator at each $t_i = i\Delta t$,

$$\mathcal{L}_{gh}(t_i, z_{t_i}, y, x) = \left\|[\alpha(t_i) \cdot y - (1 - \alpha(t_i)) \cdot G(z_{t_0}, x)] - G(z_{t_i}, x)\right\|, \tag{10}$$

where $i = 0, 1, \ldots, N$, and $\alpha(\cdot)$ is a monotonic function under the conditions $\alpha(0) = 0$ and $\alpha(T) = T$. Another perspective for the aforementioned procedure is that the *volume element* $\epsilon$ of $\mathcal{M}_y$ can be obtained as $\epsilon = \sqrt{\left|\det(\mathbf{M})\right|}dz$. Therefore, $\det(\mathbf{M})$ is a measure of the distortion in $\mathcal{M}_y$ with respect to $\mathcal{M}_z$ and geodesics prefer to avoid regions with high distortions. The procedure explained so far encourages a bi-lipschitz mapping as in Eq. 7 (proof in App. D), and satisfies **R1**. Further, we show that the enforced bijective mapping removes the loss mismatch between the adversarial and reconstruction losses, hence, improves the visual quality (see Fig. 3 and App. F).

According to **R2**, the proposed training mechanism should map Euclidean paths on $\mathcal{M}_z$ to geodesics on $\mathcal{M}_y$. Therefore, we move $z$ along Euclidean paths when minimizing $\mathcal{L}_{gh}$, which also ensures that $\mathcal{M}_z \subseteq \mathbb{R}^k$. Furthermore, we constrain $\dot{z}$ to be a constant for simplicity. Since we ensure that the distortion of $\mathcal{M}_y$ along the paths of $z$ are minimum, in practice, it can be observed that the Euclidean paths on the latent space are approximately matched to the geodesics on the output manifold (Fig. 6).

---

[2] $V$ depends on $p$ and hence the dependency of the mapping $\gamma_{\mathcal{M}_p}$ on $p$ does need to be explicitly denoted.

Further, let $\gamma_V(t)$ be a geodesic curve with an initial velocity $V$. Then, it can be shown,

$$\gamma_{cV}(t) = \gamma_V(ct), \qquad (11)$$

where $c$ is a constant (proof in App. E). This is an important result, since it immediately follows that $\left\| \dot{z}_{t_0}^1 \right\| > \left\| \dot{z}_{t_0}^2 \right\| \implies L(\gamma_{\dot{z}^1}(T)) > L(\gamma_{\dot{z}^2}(T))$. Following these intuitions, we define $\dot{z} = \nabla_z \left\| y - G(z_{t_0}) \right\|$. This yields an interesting advantage, i.e., $\|\dot{z}\|$ (hence $L(\gamma_{\dot{z}}(T))$) tends to be large for high $\left\| y - G(z_{t_0}) \right\|$, which corresponds to **R3**.

### 3.2 Encouraging the local bijective conditions

The approach described in Sec. 3.1 encourages a global bi-lipschitz mapping between $\mathcal{M}_y$ and $\mathcal{M}_z$. However, we practically observed that imposing bijective conditions in local neighborhoods in conjunction leads to improved performance. Thus, we enforce a dense bijective mapping between $\mathcal{M}_y$ and $\mathcal{M}_z$ near $\gamma_{\mathcal{M}_y}(T)$. Let $z_T$ and $y$ be the latent code at $\gamma_{\mathcal{M}_y}(T)$ and the ground truth, respectively. We generate two random sets $\tilde{\mathcal{Z}}$ and $\tilde{\mathcal{Y}}$ using the distribution,

$$\tilde{\mathcal{Z}} = \mathcal{N}(z_T; \epsilon_2) \quad \text{and} \quad \tilde{\mathcal{Y}} = \Psi(y), \qquad (12)$$

where $\Psi(\cdot)$ applies random perturbations such as brightness, contrast and small noise, and $0 < \epsilon_2 < 1$. One trivial method to ensure that a bijective mapping exists is to apply a loss function $\sum \|y_i - G(z_i)\|$, $\forall z_i \in \tilde{\mathcal{Z}}, y_i \in \tilde{\mathcal{Y}}$ to update the generator. However, we empirically observed that the above loss function unnecessarily applies a hard binding between the perturbations and the generated data. Therefore, we minimize the KL-distance between $G$ and $\tilde{\mathcal{Y}}$ up to second order moments. One possible way to achieve this is to model each pixel as a univariate distribution (App G). However in this case, since the generator cannot capture the correlations between different spatial locations, unwanted artifacts appear on the generated data. Therefore, we treat $G$ and $\tilde{\mathcal{Y}}$ as $M$-dimensional multivariate distributions ($M$ = image height $\times$ image width). Then, the KL-distance between the distributions up to the second order of moments can be calculated using the following equation,

$$\mathcal{L}_{lh}(y, z, x) = \frac{1}{2} \left[ \log \frac{|\Sigma_{G*}|}{|\Sigma_{\tilde{\mathcal{Y}}}|} - M + \mathrm{tr}(\Sigma_G^{-1} \Sigma_{\tilde{\mathcal{Y}}}) + (\mu_G - \mu_{\tilde{\mathcal{Y}}})^T \Sigma_G^{-1} (\mu_G - \mu_{\tilde{\mathcal{Y}}}) \right], \qquad (13)$$

where $\Sigma$ and $\mu$ denote the correlation matrices and the means (App. H). However, using the above loss (Eq. 13) in its original form yields practical obstacles: for instance, the correlation matrices have the dimension $M \times M$, which is infeasible to handle. Therefore, following [27], we use a random projection matrix $R^{M \times h}; h \ll M$ to project the images to a $h$-dimensional space, where $R_{i,j} \sim p(x)$; $p(\sqrt{3}) = \frac{1}{6}, p(0) = \frac{2}{3}, p(-\sqrt{3}) = \frac{1}{6}$ (we empirically justify this reduction method using an ablation study in Sec. 4). Moreover, numerically calculating $|\Sigma|$ and $\Sigma^{-1}$ causes unstable gradients which hinders the generator optimization. We address this issue by adapting the approximation technique proposed in [28]:

---

**Algorithm 1:** Training algorithm

sample inputs $\{x_1, x_2, ..., x_J\} \sim \mathcal{X}$;
sample outputs $\{y_1, y_2, ..., y_J\} \sim \mathcal{Y}$;
**for** $k$ *epochs* **do**
    **for** $x$ *in* $\chi$ **do**
        $z \sim \mathcal{B}_r^k$   //Sample $z$ from $k$-ball with a small radius $r$
        $V \leftarrow \nabla_z \|y - G(z_{t_0})\|$
        $t \leftarrow 0$
        **for** $T$ *steps* **do**
            sample noise: $e \sim \mathcal{N}(0, \epsilon_1); \epsilon_1 \ll 1$
            update $G$: $\nabla_w \mathcal{L}_{gh}(y, z, x, t)$
            update $z$: $z \leftarrow z + \eta V + e$
            update $t$: $t \leftarrow t + 1$
        update $G$:
        $\nabla_w [\mathcal{L}_{lh}(y, z, x) + \mathcal{L}_R(y, z, x) + \mathcal{L}_{adv}(y, z, x)]$

---

$$\log(|\Sigma|) \approx -\sum_{i=1}^{N} \frac{\mathrm{tr}(C^i)}{i}, \qquad (14)$$

where $C = \mathbf{I} - \Sigma$. Further, $\Sigma^{-1}$ can be calculated as,

$$V_{i+1} = V_i(3\mathbf{I} - \Sigma V_i(3\mathbf{I} - \Sigma V_n)), i = 1, 2, \ldots, N, \qquad (15)$$

where Li *et al.* [29] proved that $V_i \to \Sigma^{-1}$ as $i \to \infty$, for a suitable approximation of $V_0$. They further showed that a suitable approximation should be $V_0 = \alpha \Sigma^T, 0 < \alpha < 2/\rho(\Sigma\Sigma^T)$, where $\rho(\cdot)$ is the spectral radius. Our final loss function $\mathcal{L}_{total}$ consists of four loss components:

$$\mathcal{L}_{total} = \beta_0 \mathcal{L}_{gh} + \beta_1 \mathcal{L}_{lh} + \beta_2 \mathcal{L}_r + \beta_3 \mathcal{L}_{adv}, \qquad (16)$$

where $\beta_0 \ldots \beta_3$ are constant weights learned via cross-validation. Further, $L_{lh}$ is estimated per mini-batch. Algorithm 1 shows overall training.

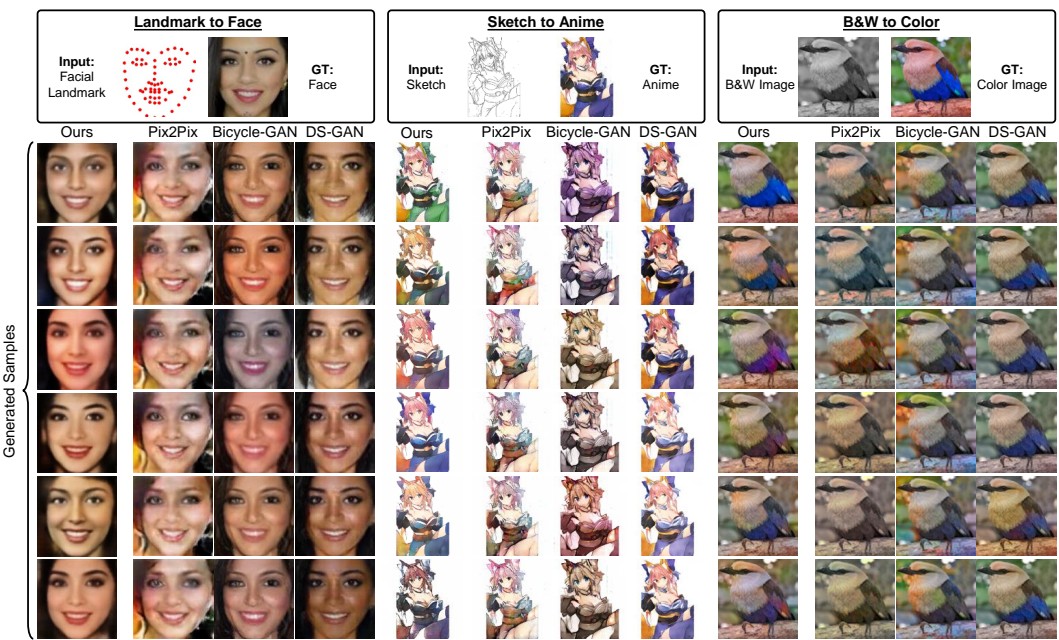

Figure 2: **Qualitative comparison with state-of-the-art cGANs on three I2I translation tasks.** We compare our model with the baseline Pix2Pix [2], Bicycle-GAN [20] and DS-GAN[18]. It can be seen that samples generated by our model are clearly more diverse (e.g., color and subtle structural variation) and realistic (e.g., shape and color) compared to other models in all tasks. Note that our model has the same architecture as Pix2Pix.

| Method | facades2photo | | sat2map | | edges2shoes | | edges2bags | | sketch2anime | | BW2color | | lm2faces | | hog2faces | | night2day | |
|---|---|---|---|---|---|---|---|---|---|---|---|---|---|---|---|---|---|---|
| | LPIPS | FID | LPIPS | FID | LPIPS | FID | LPIPS | FID | LPIPS | FID | LPIPS | FID | LPIPS | FID | LPIPS | FID | LPIPS | FID |
| Bicycle-GAN [20] | 0.142 | 58.21 | 0.109 | 54.21 | 0.139 | 21.49 | **0.184** | 22.33 | 0.026 | 73.33 | 0.008 | 78.13 | 0.125 | 72.93 | 0.065 | 98.208 | **0.103** | **120.63** |
| DS-GAN [18] | 0.181 | 59.43 | 0.128 | 48.13 | 0.126 | 27.44 | 0.113 | 26.66 | 0.006 | 67.41 | 0.012 | 71.56 | 0.168 | 88.31 | 0.061 | 92.14 | 0.101 | 137.9 |
| MR-GAN [4] | 0.108 | 110.31 | 0.091 | 108.34 | -* | -* | -* | -* | -* | -* | 0.015 | 113.46 | 0.182 | 108.72 | 0.138 | 155.31 | 0.098 | 140.51 |
| CGML [5] | **0.191** | **46.2** | 0.143 | **42.11** | 0.13 | **20.38** | 0.19 | 20.43 | 0.05 | 61.40 | 0.092 | **51.4** | 0.190 | 73.40 | 0.141 | 51.33 | 0.100 | 127.8 |
| Baseline (P2P) | 0.011 | 92.06 | 0.014 | 88.33 | 0.016 | 34.50 | 0.012 | 32.11 | 0.001 | 93.47 | 0.002 | 97.14 | 0.009 | 121.69 | 0.021 | 151.4 | 0.008 | 157.3 |
| Ours(P2P Geo) | 0.148 | 63.27 | **0.154** | 59.41 | **0.141** | 20.48 | 0.167 | **19.31** | **0.086** | **56.11** | **0.092** | 61.33 | **0.197** | **67.82** | **0.156** | **45.31** | 0.101 | 131.8 |

Table 1: **Quantitative comparison with the *state-of-the-art* on 9 (nine) challenging datasets.** -* denotes the cases where we were not able to make the models converge. A higher LPIPS similarity score means more diversity and lower FID score signifies more realism in the generated samples. Our approach gives consistent improvements over the baseline.

# 4  Experiments

In this section, we demonstrate the effectiveness of the proposed training scheme using qualitative and quantitative experiments. First, we illustrate the generalizability of our method by comparing against the state-of-the-art methods across a diverse set of image-to-image translation tasks. Then, we explore the practical implications of geometrically structuring the latent manifold. Finally, we conduct an ablation study to compare the effects of the empirical choices we made in Sec. 3. In all the experiments, we use Pix2Pix [2] as our model architecture, and use the same model trained using the traditional training approach as the main baseline. We use the official implementation of other comparable methods to benchmark their performance against ours. For a fair comparison, we use their pre-trained models wherever available, otherwise train their model from scratch, strictly following the authors' instructions to the best of our ability. For further details on the datasets and hyper-parameter settings, see App. I.

**Image-to-image translation:** We compare our method against state-of-the-art models that focus on multimodal image-to-image translation. Fig. 2 shows the qualitative results on *landmarks → faces*, *sketch → anime* and *BW → color*. As evident, our training mechanism increases the diversity and the visual quality of the baseline P2P model significantly, and shows better performance compared to other models. Fig. 3 shows qualitative comparison against the baseline. Table 1 depicts the quantitative results. As shown, our model exhibits a higher diversity and a higher realism on multiple

datasets. In all the cases, we outperform our baseline by a significant margin. Fig. 4 compares color distribution in *BW2color* task.

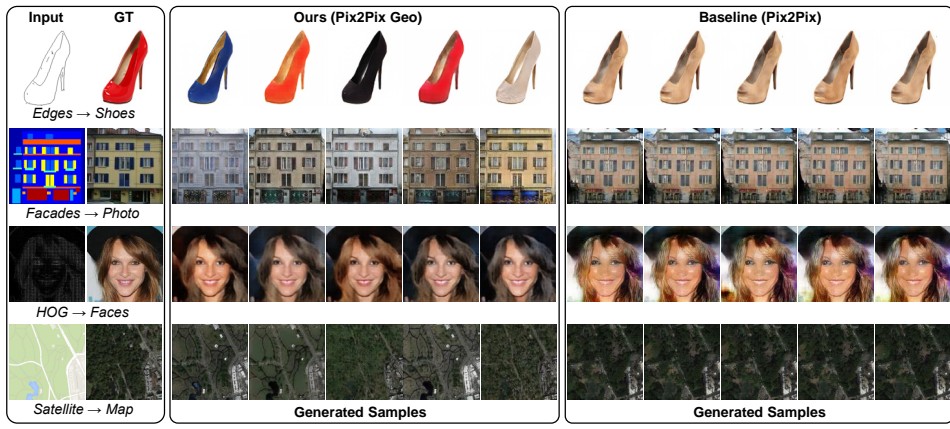

Figure 3: **Qualitative comparisons with baseline Pix2Pix [2] model.** Our proposed model consistently generates diverse and realistic samples compared to its baseline Pix2Pix model.

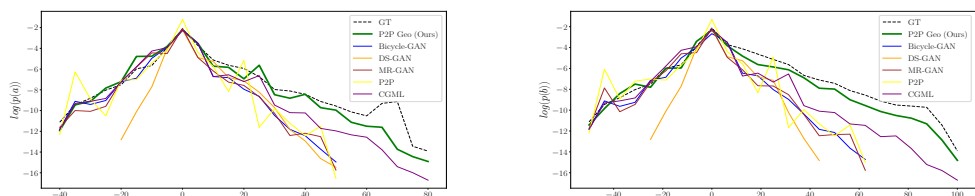

Figure 4: **Colour distribution comparison on $BW \rightarrow$ color dataset.** *left: a*-plane and *right: b*-plane in Lab color space. Our model exhibits the closest color distribution compared to the ground truth. Furthermore, our model is able to generate rare colors which implies more diverse colorization.

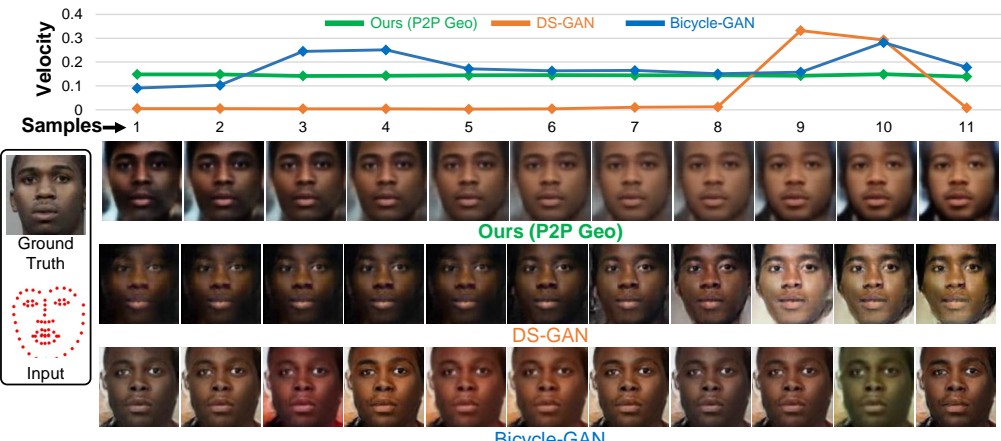

Figure 5: **A visual example of interpolation along an Euclidean shortest path on the latent manifold.** *Top row:* the velocity $V = \sqrt{\dot{z}\mathbf{M}\dot{z}}$ change on $\mathcal{M}_y$ across the samples. *Bottom three rows:* the corresponding interpolated samples in Bicycle-GAN, DS-GAN, and P2P Geo (Ours). As evident, our model exhibits a smooth interpolation along with an approximately constant velocity on $\mathcal{M}_y$ compared to the other networks, implying that our model indeed tends to move along geodesics. The total standard deviations of the $V$ for 100 random interpolations for Bicycle-GAN, DS-GAN, and P2P Geo (Ours) are 0.056 0.067, and 0.011, respectively.

**Geometrical interpretations:** A key implication of our training scheme is that the Euclidean shortest paths on $\mathcal{M}_z$ map to geodesics on $\mathcal{M}_y$, which preserves better structure. We conduct an experiment to empirically validate the aforementioned attribute. First, we travel along Euclidean paths on $\mathcal{M}_z$ and measure the corresponding curve length $L_E$ on the data manifold. Second, we calculate the actual

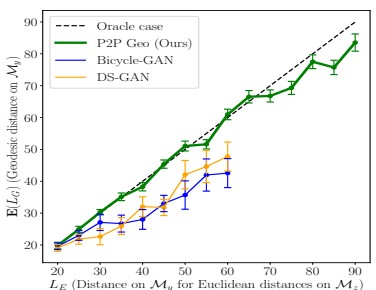

Figure 6: **Euclidean path vs. geodesic comparison.** We travel along a Euclidean shortest path on $\mathcal{M}_z$ and measure the corresponding curve distance $L_G$ on $\mathcal{M}_z$ (*lm2faces*). Then, we traverse between the same two points along the numerically calculated geodesic and measure the corresponding curve length $L_G$. $\mathbb{E}(L_G)$ vs $L_E$ is illustrated with the corresponding standard deviation obtained along 10 random paths. Our model is closer to the oracle case ($L_E = \mathbb{E}(L_G)$). We were not able to obtain distance greater than $\sim 60$ for DS-GAN and Bicyle-GAN which implies that our model generates more diverse data. Further, Pix2Pix did not produce enough diversity for this comparison.

geodesic distance $L_G$ between the same two points on $\mathcal{M}_y$ using Eq. 8 in discrete intervals. We travel in 10 random directions starting from random initial points, and obtain $L_{G_i}$ for evenly spaced $L_E \in \{10, 20, 30, \ldots 90\}$. Then, we obtain a set of the means and standard deviations of $L_G$ for the corresponding $L_E$. Fig. 6 illustrates the distribution. As evident, our model exhibits a significantly high overlap with the ideal curve, i.e., $L_E = \mathbb{E}(L_G)$ compared to DS-GAN and Bicycle-GAN.

A useful attribute of travelling along the geodesics on the output manifold ($\mathcal{M}_y$) is to obtain smooth interpolations, since the geodesics tend to avoid regions with high distortions, i.e., rapid changes. However, Euclidean shortest paths in the latent spaces ($\mathcal{M}_z$) of cGANs often do not correspond to geodesics on the $\mathcal{M}_y$. Therefore, in order to travel along geodesics, it is required to numerically obtain the geodesic paths using Eq. 8, which requires extra computation. In contrast, the proposed training method encourages the generator to map the Euclidean paths on $\mathcal{M}_z$ to geodesics on $\mathcal{M}_y$. Therefore, smooth interpolations can be obtained by traveling between two latent codes in a straight path. To evaluate this, we compare the interpolation results between Bicycle-GAN, DS-GAN and our model. Fig. 5 shows a qualitative example, along with a quantitative evaluation. As visible, our model exhibits smooth transition from the starting point to the end point. In comparison, Bicycle-GAN shows abrupt and inconsistent changes along the path. DS-GAN does not show any significant variance in the beginning and shows sudden large changes towards the end. We also quantify this comparison using the velocity on the data manifold: since the curve length on $\mathcal{M}_y$ can be calculated using Eq. 9, it is easy to see that the velocity on $\mathcal{M}_y$ can be obtained using $\sqrt{\dot{z}_i^T \mathbf{M} \dot{z}_i}$. Fig. 5 illustrates the change in the velocity, corresponding to the given qualitative examples. Our model demonstrates an approximately constant velocity (geodesics have constant velocities), while the other models show sudden velocity changes. We did not include CGML in these evaluations (App. I).

**Ablation study:** We conduct an ablation study to compare the different variants of the proposed technique. Table 2 depicts the results. First, we compare different distance functions used to calculate $\mathcal{L}_{lh}$. As expected, naive maximization of the distances between the generated samples without any constraints increases the diversity, but reduces the visual quality drastically. Further, we observed unwanted artifacts when modeling each pixel as a univariate distribution, as the model then cannot capture dependencies across spatial locations. Then, we compare different down-sampling methods that can be used for efficient calculation of the correlation matrices, where random projection performed the best. Interestingly, we observed a reduction of the visual quality when the dimension of the latent code is increased. In contrast, the diversity tends to improve with the latter. We chose $\dim(z) = 64$ as a compromise. Finally, we compare the effects of different combinations of the loss components.

| Variant type | Model | FID | LPIPS |
|---|---|---|---|
| | MMD | 66.31 | 0.188 |
| | $2^{nd}$ moment (univaritate) | 117.53 | 0.201 |
| $\mathcal{L}_{lh}$ | Maximizing distance | 132.91 | **0.232** |
| | $2^{nd}$ moment (multivariate) | **67.82** | 0.197 |
| | Mean pool | 75.41 | 0.192 |
| Downsampling | Max pool | 82.42 | 0.162 |
| | CNN | 77.93 | 0.191 |
| | Random Projection | **67.82** | **0.197** |
| | 16 | **65.32** | 0.172 |
| $\dim(z)$ | 32 | 67.11 | 0.188 |
| | 64 | 67.82 | **0.197** |
| | 128 | 82.33 | 0.166 |
| | $\mathcal{L}_l + \mathcal{L}_{adv}$ | 91.3 | 0.051 |
| Training loss | $\mathcal{L}_{gh} + \mathcal{L}_l + \mathcal{L}_{adv}$ | **63.11** | 0.151 |
| | $\mathcal{L}_{lh} + \mathcal{L}_l + \mathcal{L}_{adv}$ | 91.3 | 0.055 |
| | $\mathcal{L}_{lh} + \mathcal{L}_{gh} + \mathcal{L}_l + \mathcal{L}_{adv}$ | 67.82 | **0.197** |

Table 2: **Ablation study.** Ablation study with different variants of our model on *landmark → faces* dataset reporting FID score (lower = more realistic) and LPIPS (higher = more diverse).

**Generalizability:** To demonstrate the generalizability of the proposed algorithm across different loss functions and architectures, we employ it on three classic networks: Pathak *et al.* [3], Johnson *et al.* [30], and Ronneberger *et al.* [31]. These networks use a masked reconstruction loss with the

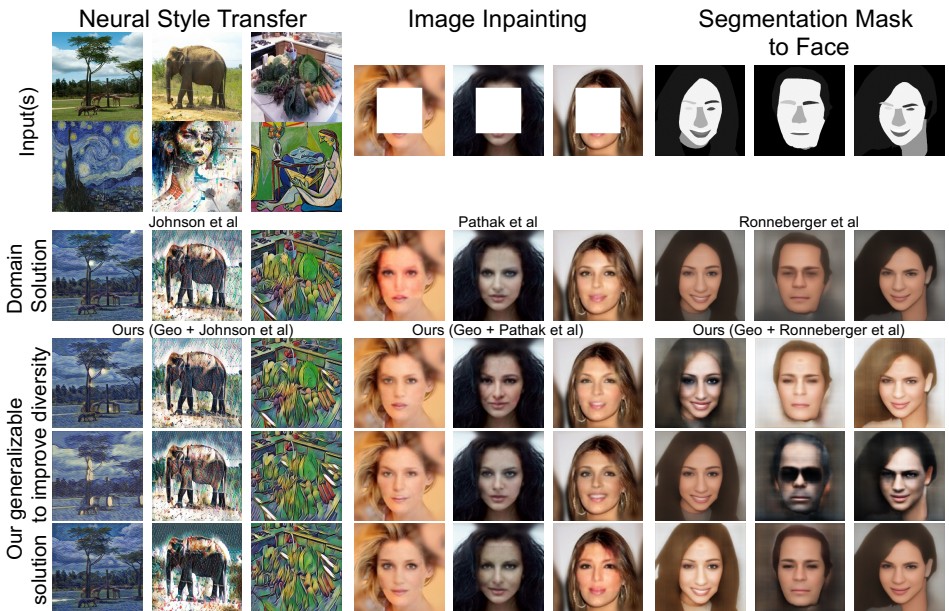

Figure 7: We apply our algorithm to three classic networks and obtain increased diversity with no architectural modifications. Note that the original networks only learn one-to-one mappings.

| Method | CelebA-HQ | | AFHQ | |
|---|---|---|---|---|
| | FID | LPIPS | FID | LPIPS |
| StarGANv2 | 13.7 | 0.452 | 16.2 | 0.450 |
| StarGANv2 (modified) | **13.1** | **0.488** | **15.7** | **0.477** |

Table 3: We apply our algorithm to StarGANv2 [32] which improves both realism and diversity.

adversarial loss, perception loss from pre-trained networks, and a reconstruction loss, respectively. Further, in the original form, these networks only learn one-to-one mappings. As depicted in Fig. 7, our algorithm increases the diversity of the models and obtains one-to-many mappings with no changes to the architecture (for fair comparison, we concatenate a latent code at the bottlenecks during both the original and proposed training). Further, since our algorithm is architecture agnostic, it can be easily injected into any suitable model. To show this, we replace the $L_{ds}$ of StarGANv2 [32] with our loss $L_{gh}$. In StarGANv2, the images are generated as a composition of two functions: the style generator ($F(\cdot)$) and the image generator ($G(\cdot, \cdot)$). We enforce a bi-lipschitz constraint between the latent space and the image generator outputs. For continuously differentiable functions $F$ and $G$, if the composition $G \circ F$ is homeomorphic, $F$ and $G$ are individually homeomorphic. Hence, the image generator output becomes homeomorphic to the style space. As shown in Table 3, our approach can improve the performance of StarGANv2, without any modifications to the architecture.

## 5 Conclusion

We show that the cGANs, in their basic form, suffer from significant drawbacks in-terms of diversity and realism. We propose a novel training algorithm that can increase both realism and the diversity of the outputs that are generated by cGANs while preserving the structure of the latent manifold. To this end, we enforce a bi-lipschitz mapping between the latent and generated output manifolds while encouraging Euclidean shortest paths on the latent manifold to be mapped to the geodesics on the generated manifold. We establish the necessary theoretical foundation and demonstrate the effectiveness of the proposed algorithm at a practical level, using a diverse set of image-to-image translation tasks, where our model achieves compelling results.

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
