# Appendix

## A  Funding transparency statement

Funding in direct support of this work: Data61 PhD Scholarship (Postgraduate Research Scholarship (675/2014)) and Data61 Top-up Scholarship (Postgraduate Research Scholarship (675/2014)).


## C  Geodesics are uniquely defined by an initial velocity and a point on the manifold.

Although this is a standard result in geometry, we state the proof here for completeness. Let $\mathcal{M}$ be a manifold and $\gamma : I \to \mathcal{M}$ be a geodesic satisfying $\gamma(t_0) = p$, $\dot{\gamma}(t_0) = V$, where $p \in \mathcal{M}$, $V \in T_p\mathcal{M}$

and $I \subset \mathbb{R}$. $T_p\mathcal{M}$ is the tangent bundle of $\mathcal{M}$. Let us choose coordinates $(x^i)$ on a neighborhood $U$ of $p$, s.t. $\gamma(t) = (x^1(t), x^2(t), \ldots, x^n(t))$. For $\gamma$ to be a geodesic it should satisfy the condition,

$$\ddot{x}^k + \dot{x}^i(t)\dot{x}^j(t)\Gamma^k_{ij}(x(t)) = 0, \tag{17}$$

where Eq. 17 is written using Einstein summation. Here, $\Gamma$ are Christoffel symbols that are functions of the Riemannian metric. Eq. 17 can be interpreted as a second-order system of ordinary differential equations for the functions $x^i(t)$. Using auxiliary variables $v_i = \dot{x}^i$, it can be converted to an equivalent first-order system as,

$$\dot{x}^k(t) = v^k(t), \tag{18}$$

$$\dot{v}^k(t) = -v^i(t)v^j(t)\Gamma^k_{ij}(x(t)). \tag{19}$$

On the other hand, existence and uniqueness theorems for first-order ODEs ensure that for any $(p, V) \in U \times \mathbb{R}^n$, there exists a unique solution $\eta : (t_0 - \epsilon, t_0 + \epsilon) \to U \times \mathbb{R}^n$, where $\epsilon > 0$, satisfying the initial condition $\eta(t_0) = (p, V)$.

Now, let us define two geodesics, $\gamma, \beta : I \to \mathcal{M}$ in an open interval with $\gamma(t_0) = \beta(t_0)$ and $\dot{\gamma}(t_0) = \dot{\beta}(t_0)$. By the above mentioned uniqueness theorem, they agree on some neighborhood of $t_0$. Let $\alpha$ be the supremum of numbers $b$ s.t. they agree on $[t_0, b]$. If $\alpha \in I$, then using continuity it can be seen that, $\gamma(\alpha) = \beta(\alpha)$ and $\dot{\gamma}(\alpha) = \dot{\beta}(\alpha)$. Then, by applying local uniqueness in a neighborhood of $\alpha$, the curves agree on a slightly larger interval, which is contradiction. Hence, arguing the similarity to the left of $t_0$, it can be seen that the curves agree on all $I$.

## D    Proof for Eq. 7

The loss $\mathcal{L}_{gh}$ in Sec. 3.1 forces $G(z)$ to be smooth and $\det(\frac{\partial(G)}{dz}) > 0$, hence, $\det(\mathbf{G}) > 0$. Let $T(\cdot)$ denote the unit tangent bundle of a given manifold. Then, the map $df : T\mathcal{M}_z \to T\mathcal{M}_y$ is also smooth. Therefore, the function $h(p) = |df(p)|$, $p \in T\mathcal{M}_z$ is continuous too. Let $1/C$ and $K$ denote its minimum and maximum, respectively. Therefore, for every unit speed piecewise-smooth path $\gamma : [a, b] \to \mathcal{M}_z$, the length of its image in $\mathcal{M}_y$ is,

$$L(G \circ \gamma) = \int_a^b \left\| \frac{\partial(G \circ \gamma)}{dt} \right\| dt. \tag{20}$$

Further,

$$\frac{1}{C} \int_a^b \left\| \frac{\partial\gamma}{dt} \right\| dt < L(G \circ \gamma) < K \int_a^b \left\| \frac{\partial\gamma}{dt} \right\| dt. \tag{21}$$

If $C < K$,

$$\frac{1}{K} \int_a^b \left\| \frac{\partial\gamma}{dt} \right\| dt < L(G \circ \gamma) < K \int_a^b \left\| \frac{\partial\gamma}{dt} \right\| dt. \tag{22}$$

On the contrary, if $C \geq K$,

$$\frac{1}{C} \int_a^b \left\| \frac{\partial\gamma}{dt} \right\| \leq L_{\mathcal{M}_y}(G \circ \gamma) \leq C \int_a^b \left\| \frac{\partial\gamma}{dt} \right\|. \implies \frac{1}{C} L_{\mathcal{M}_z}(\gamma) \leq L_{\mathcal{M}_y}(\gamma) \leq C L_{\mathcal{M}_z}(\gamma). \tag{23}$$

Since the geodesic distances are length minimizing curves on $\mathcal{M}_y$ and $\mathcal{M}_z$, it follows that,

$$\frac{1}{C} d_{\mathcal{M}_z}(z^p, z^q) \leq d_{\mathcal{M}_y}(\phi^{-1}(z^p), \phi^{-1}(z^q)) \leq C d_{\mathcal{M}_z}(z^p, z^q), \tag{24}$$

where, $d(\cdot, \cdot)$ are the geodesic distances and $C$ is a constant.

# E Proof for Eq. 11

Consider a geodesic $\gamma_V : I \to \mathcal{M}$, defined in an open interval $I \subset \mathbb{R}$, with an initial velocity $V \in T\mathcal{M}$. Let us also define a curve $\tilde{\gamma}(t) = \gamma_V(ct)$. Then, $\tilde{\gamma}(0) = \gamma_V(0) = p \in \mathcal{M}$. Writing $\gamma_V(t) = (\gamma^1(t), \gamma^2(t), \ldots, \gamma^n(t))$ in local coordinates,

$$\dot{\tilde{\gamma}}(t) = \frac{d}{dt}\gamma_V^i(ct) = c\dot{\gamma}_V^i(ct). \tag{25}$$

Further, it follows that $\dot{\tilde{\gamma}} = c\dot{\gamma}(0) = cV$.

Next, let $D_t$ and $\tilde{D}_t$ denote the covariant differentiation operators along $\gamma_V$ and $\tilde{\gamma}$, respectively. Then,

$$\tilde{D}_t\dot{\tilde{\gamma}}(t) = [\frac{d}{dt}\dot{\tilde{\gamma}}^k(t) + \Gamma_{ij}^k(\tilde{\gamma}(t))\dot{\tilde{\gamma}}^i(t)]\partial_k \tag{26}$$

$$= (c^2\ddot{\gamma}^k(ct) + c^2\Gamma_{ij}^k(\gamma_V(ct))\dot{\gamma}_V^i(ct)\dot{\gamma}^j(ct))\partial_k \tag{27}$$

$$c^2 D_t\dot{\gamma}(ct) = 0. \tag{28}$$

Hence, $\tilde{\gamma}$ is a geodesic, and therefore, $\tilde{\gamma} = \gamma_{cV}$.

# F Removing the loss mismatch using the proposed method.

The set of optimal generator $G^*$ for the adversarial loss can be formulated as,

$$G^* = \underset{G}{\operatorname{argmin}}\Big(\mathbf{JSD}\big[p_g(\bar{y})\|p_d(y)\big]\Big), \tag{29}$$

where $\mathbf{JSD}$ is the Jensen–Shannon divergence, $y$ is the ground-truth and $\bar{y} = G(z)$ is the generated output.

Now, let us consider the expected $\ell_1$ loss, $\mathbb{E}_{y,z}|y - \bar{y}(z)|$. Then,

$$\mathbb{E}_{y,z}|y - \bar{y}(z)| = \int_{-\infty}^{\infty}\int_{-\infty}^{\infty}|y - \bar{y}(z)|\,p(y)p(z|y)dzdy. \tag{30}$$

To find the minimum of the above, we find the value where the subderivative of the $\bar{y}(z)$ equals to zero as,

$$\frac{d}{d\bar{y}}[\int_{-\infty}^{\infty}\int_{-\infty}^{\infty}|y - \bar{y}(z)|\,p(y)p(z|y)dydz] = \int_{-\infty}^{\infty}\int_{-\infty}^{\infty}-\operatorname{sign}(y - \bar{y}(z))p(y)p(z|y)dzdy = 0. \tag{31}$$

$$\int_{-\infty}^{\bar{y}}\int_{-\infty}^{\infty}-\operatorname{sign}(y - \bar{y}(z))p(y)p(z|y)dzdy + \int_{\bar{y}}^{\infty}\int_{-\infty}^{\infty}-\operatorname{sign}(y - \bar{y}(z))p(y)p(z|y)dzdy = 0. \tag{32}$$

$$\int_{-\infty}^{\bar{y}}\int_{-\infty}^{\infty}p(y)p(z|y)dzdy = \int_{\bar{y}}^{\infty}\int_{-\infty}^{\infty}p(y)p(z|y)dzdy. \tag{33}$$

Since $z$ is randomly sampled, with enough iterations $p(z) = p(z|y)$. Then,

$$\int_{-\infty}^{\bar{y}}p(y)dy\int_{-\infty}^{\infty}p(z)dz = \int_{\bar{y}}^{\infty}p(y)\int_{-\infty}^{\infty}p(z)dz, \tag{34}$$

$$\int_{-\infty}^{\bar{y}}p(y)dy = \int_{\bar{y}}^{\infty}p(y)dy, \tag{35}$$

which means that the probability mass to left of $\bar{y}$ is equal to the probability mass to the right of $\bar{y}$. Therefore, $\bar{y}$ is the median of the distribution $p(y)$. Hence, unless $p_d(y)$ is unimodal with a sharp peak, the optimal generator for the $\ell_1$ loss does not equal $G^*$.

Now, consider a function $f$ such that $f(z) = y$ and $p(f(z)) = p_d$. Then, the corresponding cumulative distribution is,

$$F(y) = p(f(z) \leq y). \tag{36}$$

Therefore, $p(f(z))$ can be obtained as,

$$p(f(z)) = \frac{\partial}{\partial y_1} \cdots \frac{\partial}{\partial y_M} \int_{\{z^* \in \mathbb{R}^k \mid f(z^*) \leq f(z)\}} p(z) d^k z. \tag{37}$$

According to Eq. 37, $f$ should be differentiable almost everywhere with a positive definite $\mathbf{J}_f^T \mathbf{J}_f$, where $\mathbf{J}_f$ is the Jacobian of $f$. Recall the Rademacher theorem,

**Theorem 1**: Let $\mathcal{Z}$ be an open subset of $\mathbb{R}^k$ and $g : \mathcal{Z} \to \mathbb{R}^M$ a lipschitz function. Then, $g$ differentiable almost everywhere (with respect to the Lebesgue measure $\lambda$). That is, there is a set $E \subset \mathcal{Z}$ with $\lambda(\mathcal{Z}/E) = 0$ and such that for every $z \in E$ there is a linear function $L_z : \mathbb{R}^k \to \mathbb{R}^M$ with

$$\lim_{z^* \to z} \frac{g(z) - g(z^*) - L_z(z^* - z)}{|z^* - z|} = 0. \tag{38}$$

Recall that our loss function enforce a bilipschitz mapping between the manifolds with a positive definite metric tensor, hence, $G^{-1}$ and $G$ is differentiable almost everywhere. That is, given enough flexibility, $G$ converges to $f$ almost surely, i.e., $\mathbf{JSD}[p_g(\bar{y})||p_d(y)] \approx 0$. Hence, our adversarial loss and the other loss components are not contradictory.

## G  Univariate distributions

Minimizing the information loss between two distributions can be interpreted as minimizing the Kullback–Leibler (KL) distance between the two distributions. KL-distance between two distribution is defined as,

$$KL(P||Q) = \int p(x) \log\left[\frac{p(x)}{q(x)}\right] dx. \tag{39}$$

If we approximate an arbitrary density $Q$ in $\mathbb{R}^n$ with a Gaussian distribution, it can be shown that the parameters which minimize the KL-distance between $Q$ and a given density $P$ are exactly the same as minimizing the distance between $P$ and $Q$ up to the second moment. Therefore, we approximate $P$ and $Q$ with Gaussian distributions and minimize the KL distance between them.

Now, consider two Gaussian distributions, $P$ and $Q$.

$$
\begin{aligned}
KL(P||Q) &= \int \left[ \log(P(x)) - \log(Q(x)) \right] P(x) dx \\
&= \int \left[ -\frac{1}{2}\log(2\pi) - \log(\sigma_P) - \frac{1}{2}(\frac{x - \mu_P}{\sigma_P})^2 \right. \\
&\quad \left. + \frac{1}{2}\log(2\pi) + \log(\sigma_Q) + \frac{1}{2}(\frac{x - \mu_Q}{\sigma_Q})^2 \right] \frac{1}{\sqrt{2\pi}\sigma_P} \exp\left[ -\frac{1}{2}(\frac{x - \mu_P}{\sigma_P})^2 \right] dx \\
&= \int \left[ \log(\frac{\sigma_Q}{\sigma_P}) + \frac{1}{2}((\frac{x - \mu_Q}{\sigma_Q})^2 - (\frac{x - \mu_P}{\sigma_P})^2) \right] \frac{1}{\sqrt{2\pi}\sigma_P} \exp\left[ -\frac{1}{2}(\frac{x - \mu_P}{\sigma_P})^2 \right] dx \quad (40) \\
&= \mathbb{E}_P\left[ \log(\frac{\sigma_Q}{\sigma_P}) + \frac{1}{2}((\frac{x - \mu_Q}{\sigma_Q})^2 - (\frac{x - \mu_P}{\sigma_P})^2) \right] \\
&= \log(\frac{\sigma_Q}{\sigma_P}) + \frac{1}{2\sigma_Q^2}\mathbb{E}_P[(x - \mu_Q)^2] - \frac{1}{2} \\
&= \log(\frac{\sigma_Q}{\sigma_P}) + \frac{\sigma_P^2 + (\mu_P - \mu_Q)^2}{2\sigma_Q^2} - \frac{1}{2}.
\end{aligned}
$$

# H  Multivariate distribution

Consider two Gaussian distributions, $P$ and $Q$ in $\mathbb{R}^n$,

$$P(x) = \frac{1}{(2\pi)^{n/2}\det(\Sigma_P)^{1/2}}\exp\left[-\frac{1}{2}(x-\mu_P)^T\Sigma_P^{-1}(x-\mu_P)\right], \tag{41}$$

$$Q(x) = \frac{1}{(2\pi)^{n/2}\det(\Sigma_Q)^{1/2}}\exp\left[-\frac{1}{2}(x-\mu_Q)^T\Sigma_Q^{-1}(x-\mu_Q)\right]. \tag{42}$$

KL distance between the two distributions,

$$
\begin{aligned}
KL(P\|Q) &= \mathbb{E}_P\left[\log P - \log Q\right] \\
&= \frac{1}{2}\mathbb{E}_P\left[-\log\det\Sigma_P - (x-\mu_P)^T\Sigma_P^{-1}(x-\mu_P) + \log\det\Sigma_Q + (x-\mu_Q)^T\Sigma_Q^{-1}(x-\mu_Q)\right] \\
&= \frac{1}{2}\left[\log\frac{\det\Sigma_Q}{\det\Sigma_P}\right] + \frac{1}{2}\mathbb{E}_P\left[-(x-\mu_P)^T\Sigma_P^{-1}(x-\mu_P) + (x-\mu_Q)^T\Sigma_Q^{-1}(x-\mu_Q)\right] \\
&= \frac{1}{2}\left[\log\frac{\det\Sigma_Q}{\det\Sigma_P}\right] + \frac{1}{2}\mathbb{E}_P\left[-\text{tr}(\Sigma_P^{-1}(x-\mu_P)(x-\mu_P)^T) + \text{tr}(\Sigma_Q^{-1}(x-\mu_Q)(x-\mu_Q)^T)\right] \\
&= \frac{1}{2}\left[\log\frac{\det\Sigma_Q}{\det\Sigma_P}\right] + \frac{1}{2}\mathbb{E}_P\left[-\text{tr}(\Sigma_P^{-1}\Sigma_P) + \text{tr}(\Sigma_Q^{-1}(xx^T - 2x\mu_Q^T + \mu_Q\mu_Q^T))\right] \\
&= \frac{1}{2}\left[\log\frac{\det\Sigma_Q}{\det\Sigma_P}\right] - \frac{1}{2}M + \frac{1}{2}\text{tr}(\Sigma_Q^{-1}(\Sigma_P + \mu_P\mu_P^T - 2\mu_Q\mu_P^T + \mu_Q\mu_Q^T)) \\
&= \frac{1}{2}(\left[\log\frac{\det\Sigma_Q}{\det\Sigma_P}\right] - M + \text{tr}(\Sigma_Q^{-1}\Sigma_P) + \text{tr}(\mu_P^T\Sigma_Q^{-1}\mu_P - 2\mu_P^T\Sigma_Q^{-1}\mu_Q + \mu_Q\Sigma_Q^{-1}\mu_Q)) \\
&= \frac{1}{2}(\left[\log\frac{\det\Sigma_Q}{\det\Sigma_P}\right] - M + \text{tr}(\Sigma_Q^{-1}\Sigma_P) + (\mu_Q-\mu_P)^T\Sigma_Q^{-1}(\mu_Q-\mu_P)).
\end{aligned}
\tag{43}
$$

# I  Hyper-parameters and datasets

We use 100 iterations with $\alpha = 0.1$ to calculate the inverse of matrices using Eq. 15 and 20 iterations to calculate the log determinant using Eq. 14. Further, 10 time steps are used for $\mathcal{L}_{gh}$, and $z_{t_0}$ is sampled from a $\mathcal{B}_{0.01}^{64}$. For training, we use the Adam optimizer with hyper-parameters $\beta_1 = 0.9, \beta_2 = 0.999, \epsilon = 1 \times 10^{-8}$. All the weights are initialized using a random normal distribution with 0 mean and 0.5 standard deviation. The weights of the final loss function are,

$$\mathcal{L}_{total} = 100.0\mathcal{L}_{gh} + 0.01\mathcal{L}_{lh} + 100.0\mathcal{L}_R + \mathcal{L}_{adv}, \tag{44}$$

All these values are chosen empirically. For $facades \rightarrow photo$, $map \rightarrow photo$, $edges \rightarrow shoes$, $edges \rightarrow bags$, and $night \rightarrow day$, we use the same stadard datasets used in Pix2Pix [2]. For the $landmarks \rightarrow faces$, $hog \rightarrow faces$, $BW \rightarrow color$, and $sketch \rightarrow anime$ experiments, we use the UTKFace dataset [33], CelebHQ dataset [34], STL dataset [35], and *Anime Sketch Colorization Pair* dataset [36] provided in Kaggle, respectively.

## I.1  Incompatibility of CGML with experiments that evaluate the latent structure

The inference procedure of the CGML is fundamentally different from a CGAN. The latent variables are randomly initialized at inference and then guided towards optimal latent codes through a separate path-finding expert module. As a result, unlike CGANs, the entire latent space is not considered as a low-dimensional manifold approximation of the output space. In other words, interpolation through sub-optimal areas of the latent space does not correspond to meaningful changes in the output. Therefore, we did not use CGML for experiments that evaluate the structure of the latent space.

## I.2  Experiments on the generalizability of the proposed algorithm

We utilized three networks for this experiment: Pathak *et al.* [3], Johnson *et al.* [37], and Ronneberger *et al.* [31].

**Pathak *et al.*** This model is proposed for image inpainting tasks. The model is trained by regressing to the ground truth content of the missing area. To this end, they utilize a reconstruction loss ($L_{rec}$) and an adversarial loss ($L_{Adv}$). Consider a binary mask $M$ where missing pixels are indicated by 1 and 0 otherwise. Then, $L_{rec}(x) = \left\| M \odot (x - G((1 - M) \odot x)) \right\|_2^2$, where $x$ is the input and $\odot$ is the element-wise production. $L_{adv}$ is the usual adversarial loss on the entire output. In order to apply our training algorithm, we replace $L_R$ with $L_{rec}$.

**Johnson *et al.*** The primary purpose of this network is neural style transferring, *i.e.*, given a artistic style image and an RGB image, output should construct an image where the content of the RGB image is represented using the corresponding artistic style. The model utilizes an encoder decoder mechanism and consists of four loss components: 1) feature reconstruction loss $\mathcal{L}_{fr}$, 2) style reconstruction loss $\mathcal{L}_{style}$ 3) reconstruction loss and 4) variation regularization loss $\mathcal{L}_{tv}$. The feature reconstruction loss is obtained by passing the generated and ground truth images through a pre-trained VGG-16 and calculating the $\ell_2$ loss between the corresponding feature maps. Let the output of the *relu2_2* layer of VGG-16 be denoted as $\phi(\cdot)$. Then,

$$\mathcal{L}_{fr}(y, \bar{y}) = \frac{1}{K} \left\| \phi(y) - \phi(\bar{y}) \right\|_2^2, \tag{45}$$

where K is the number of neurons in *relu2_2*.

The style reconstruction loss is similar, except that the inputs to the VGG-16 are the generated image and the style image. Let the output of the $j^{th}$ layer of VGG-16 be $\phi(\cdot)_j$. Further, assume that $\phi(\cdot)_j$ gives $C_j$ dimensional features on a $H_j \times W_j$ grid, which can be reshaped in to a $C_j \times H_j W_j$ matrix $\psi_j$. Then, $G_j(\cdot) = \psi \psi^T / (C_j H_j W_j)$ and,

$$\mathcal{L}_{style} = \left\| G_j(y) - G_j(\bar{y}) \right\|_F^2, \tag{46}$$

where $\|\cdot\|_F$ is the Frobeneus norm. While training, $\mathcal{L}_{style}$ is calculated for *relu1_2, relu2_2, relu3_3*, and *relu4_3* of the VGG-16.

Reconstruction loss is simply the pixel-wise $\ell_2$ loss. They also adapt a total variation loss to encourage spatial smoothness in the output image as,

$$\mathcal{L}_{tv}(\bar{y}) = \sum_{i,j} ((\bar{y}_{i,j+1}, -\bar{y}_{i,j})^2 + (\bar{y}_{i+1,j}, -\bar{y}_{i,j})^2). \tag{47}$$

In order to apply our training algorithm, we replace $L_{Adv}$ with $\mathcal{L}_{fr}$, $\mathcal{L}_{style}$, and $\mathcal{L}_{tv}$.

**Ronneberger *et al.*** This model was originally proposed for segmentation of RGB (medical) images and is trained with a soft-max cross-entropy loss between the predicted and target classes. However, we use a pixel-wise reconstruction loss as the objective function to allow multi-modal outputs. Further, we define the task at hand as converting segmentation maps to faces. To impose our training algorithm, we simply remove $L_{adv}$.

The above networks are designed to capture one-to-one mappings between the inputs and the outputs. Therefore, the only stochasticity in these models is the dropout. Therefore, we concatenate a latent map to the bottle-necks of the networks to improve the stochasticity. Note that simply concatenating the latent maps without our algorithm does not yield diverse outputs as the naive reconstruction losses (which exist in all of the above networks) only converge to a single output mode.

## J  Discussion on Related works

**Conditional generative modeling.** Generative modeling has shown remarkable progress since the inception of Variational Autoencoders (VAE) [38] and GANs [1]. Consequently, the conditional

counter-parts of these models have dominated the conditional generative tasks [2, 39, 40, 41, 42, 43]. However, conditional generation in multimodal spaces remain challenging, as the models need to exhibit a form of stochasticity in order to generate diverse outputs. To this end, Zhu *et al.* [20] proposed a model where they enforce a bijective mapping between the outputs and the latent spaces. Yang *et al.* [18], Mao *et al.* [19], and Lee *et al.* [4] introduced novel objective functions to increase the distance between the samples generated for different latent seeds. Chang *et al.* [44] used separate variables that can be injected at the inference to change the effects of loss components that were used during the training. In contrast, VAE based methods aim to explicitly model the latent probability distribution and at inference, diverse samples are generated using different latent seeds. However, typically, the latent posterior distribution of the VAE is approximated by a Gaussian, hence, the ability to model more complex distributions is hindered. As a solution, Maaloe *et al.* [45] suggested using auxiliary variables to hierarchically generate more complex distributions, using a Gaussian distribution as the input. Normalizing Flows [46] are similar in concept, where the aim is to generate more complex posterior distributions hierarchically. They apply a series of bijective mappings to an initial simple distribution, under the condition that the Jacobian of these mappings are easily invertible.

**Geometrical analysis of generative models.** Recent works have discovered intriguing geometrical properties of generative models [17, 16, 23]. These works apply post-train analysis on the models and confirm that Euclidean paths in the latent space do not map to geodesics on the generated manifold. In contrast, we focus on preserving these properties while training the model. In another direction, Wang *et al.* [24] introduced a loss function that forces the real and generated distributions to be matched in the topological feature space. They showed that by using this loss, the generator is able to produce images with the same structural topology as in real images. Similarly, Khrulkov *et al.* [25] proposed a novel performance metric for GANs by comparing geometrical properties of the real and generated data manifolds. Different to our work, these methods do not ensure homeomorphism between the latent and generated manifolds.

## K   Qualitative results

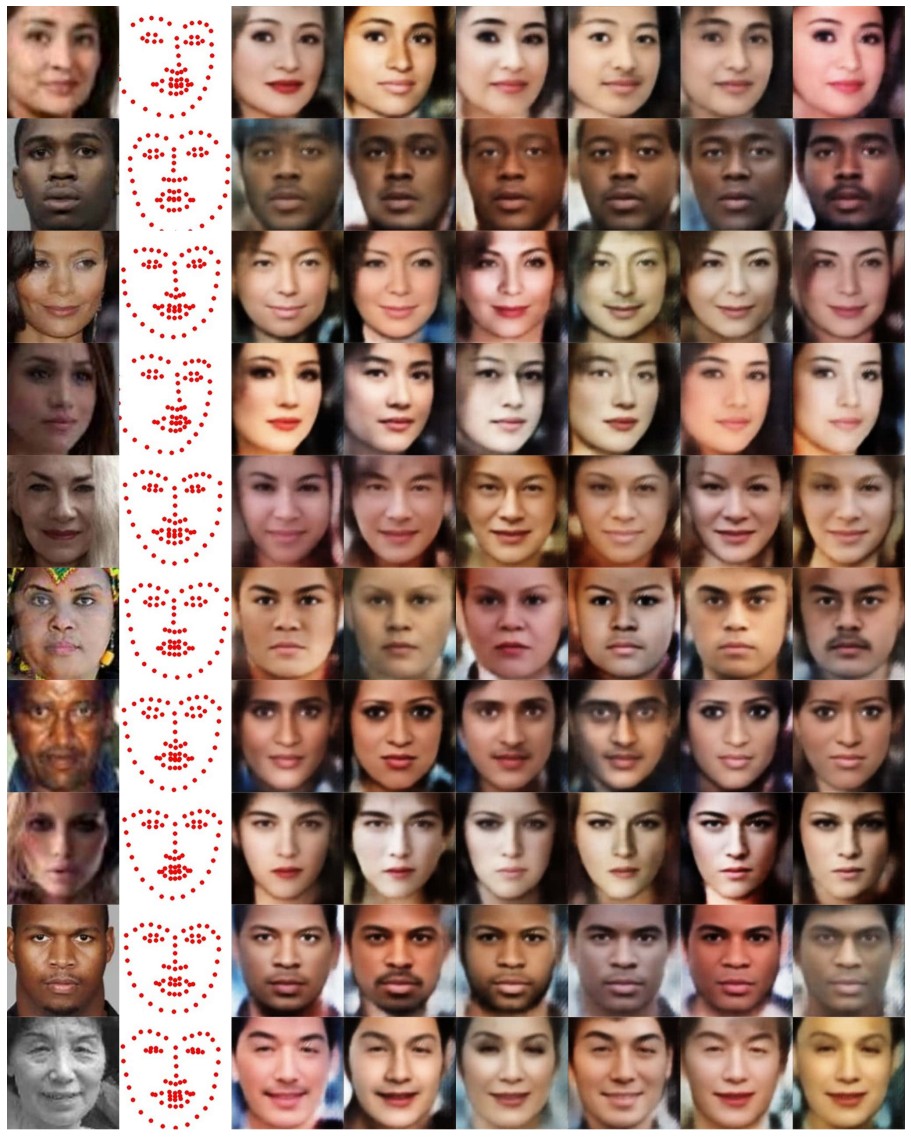

GT      Input                Generated samples

Figure 8: Qualitative results from *landmarks → faces* task.

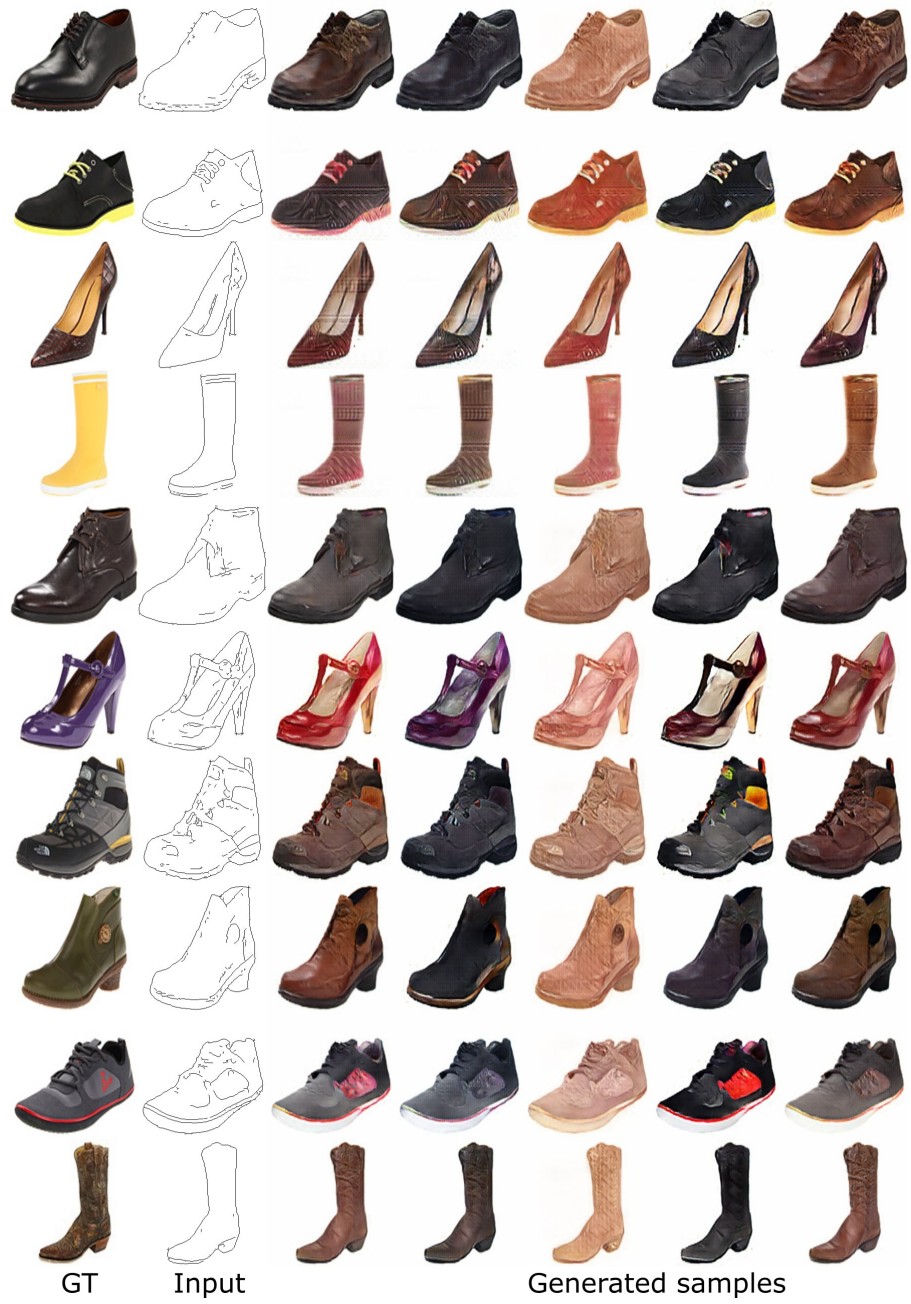

GT  Input       Generated samples

Figure 9: Qualitative results from *sketch → shoes* task.

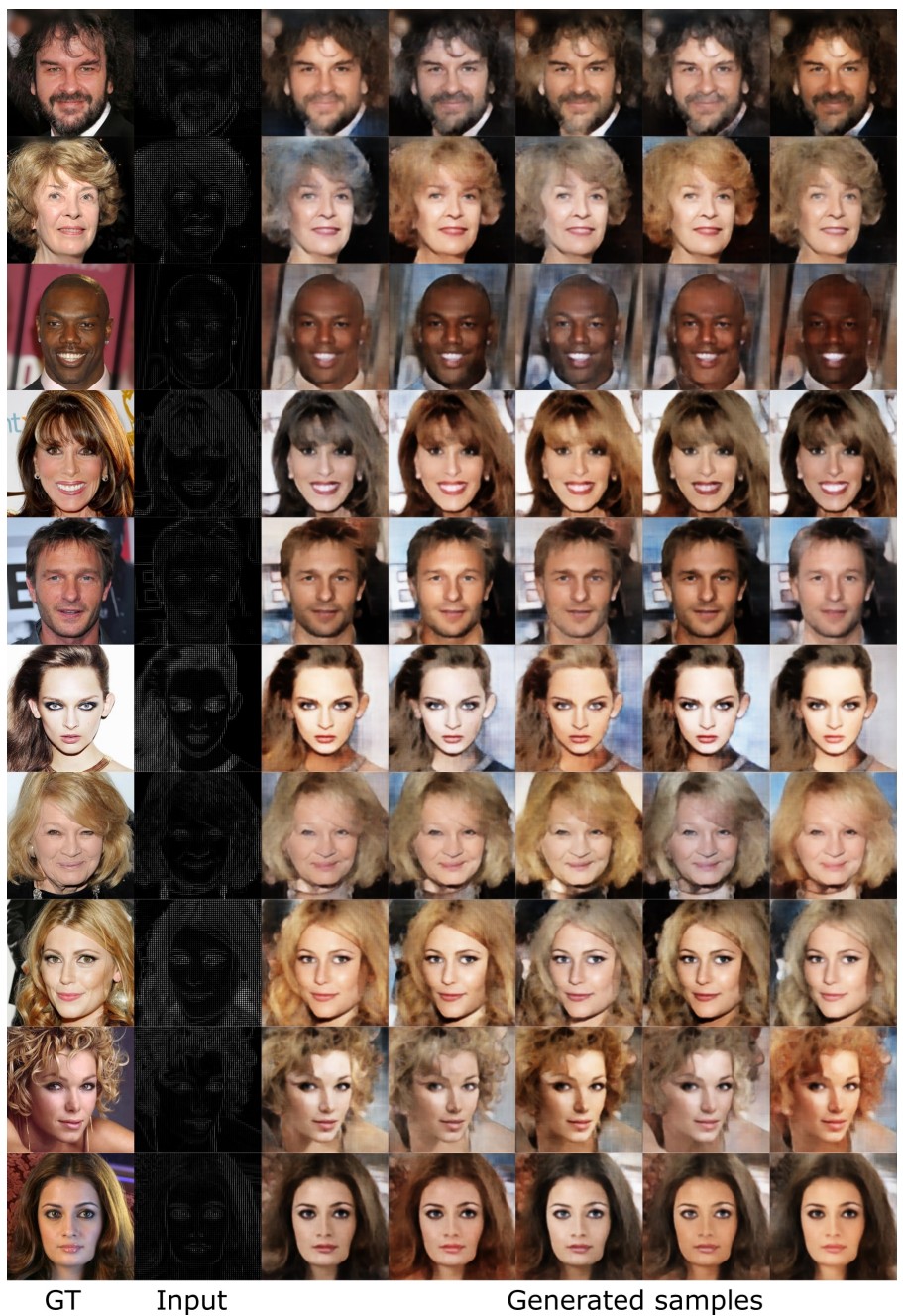

GT    Input        Generated samples

Figure 10: Qualitative results from *hog → faces* task. The diversity of the outputs are less in this task, as hog features maps are rich in information.

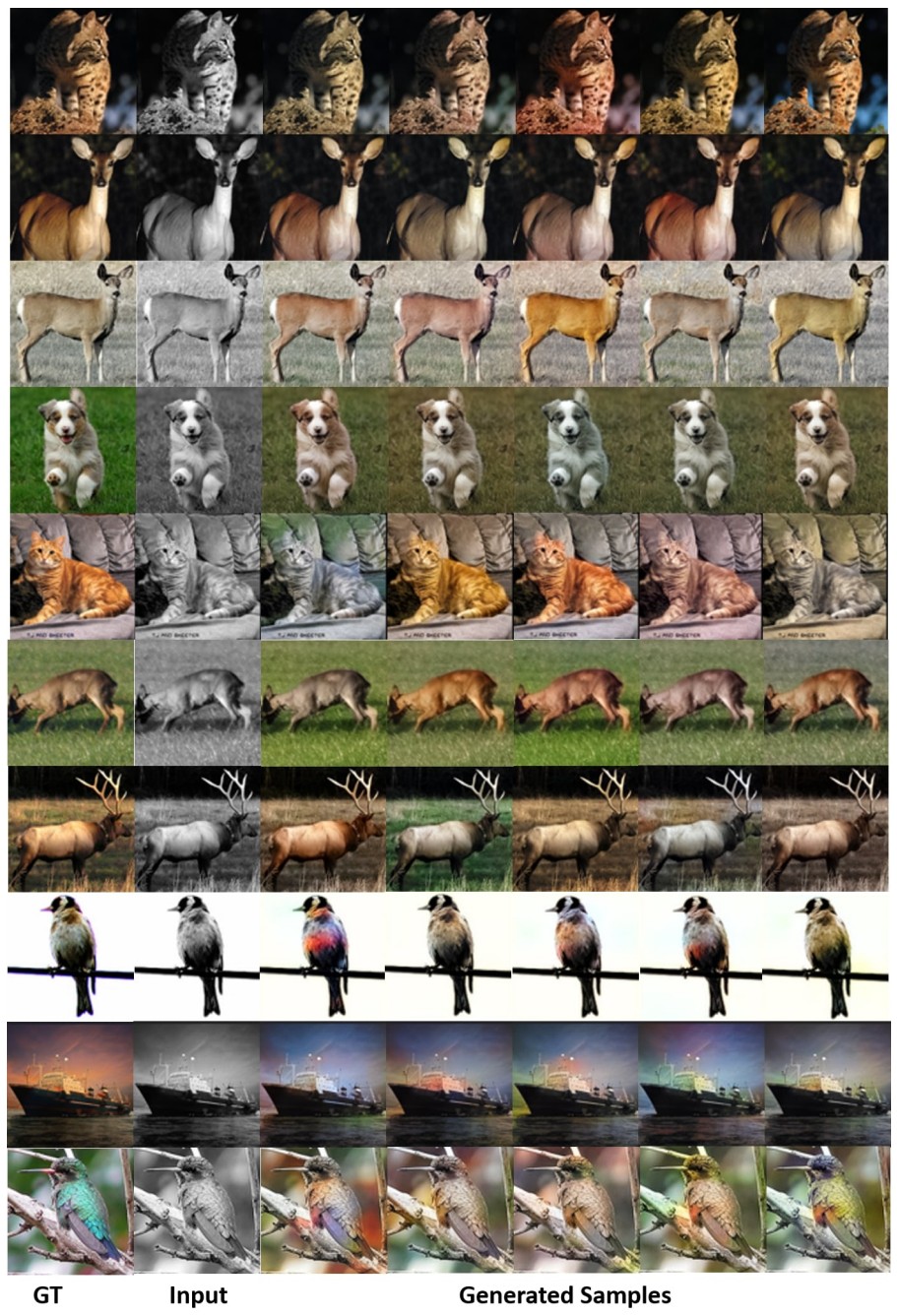

**GT**  **Input**        **Generated Samples**

Figure 11: Qualitative results from *BW → color* task.

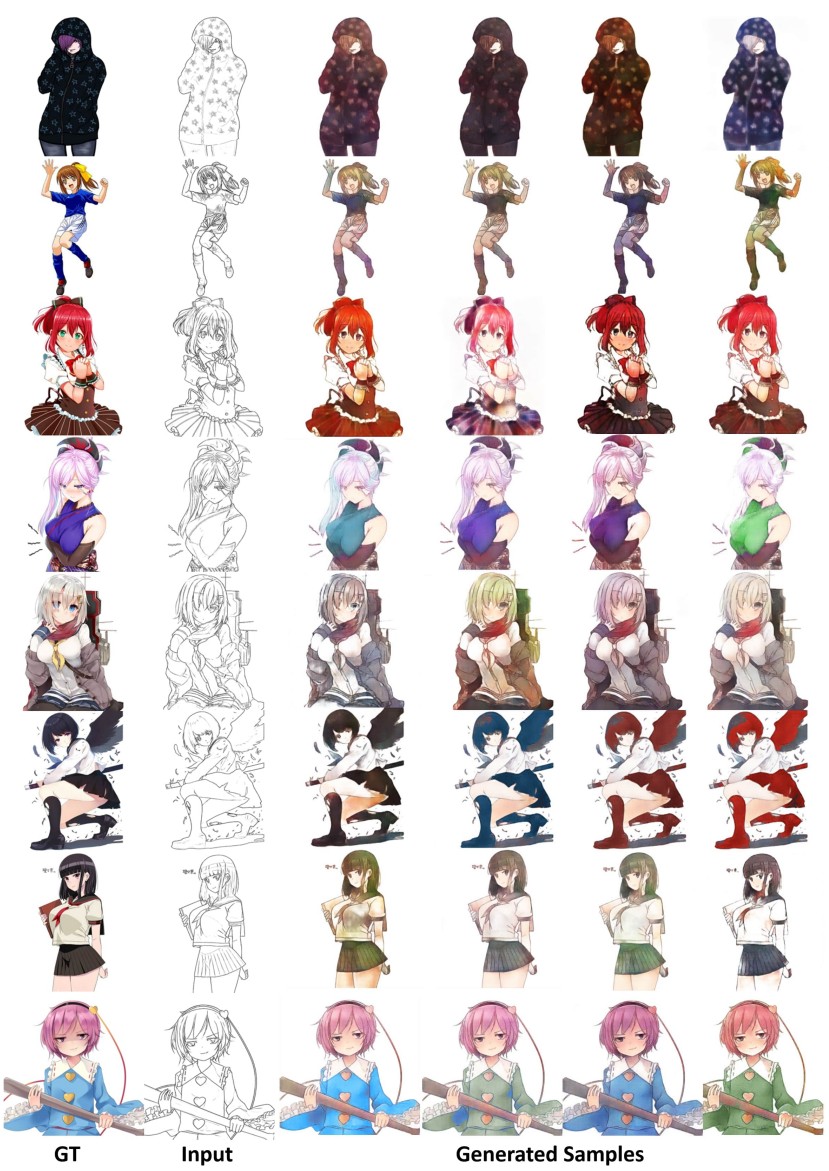

**GT**     **Input**                    **Generated Samples**

Figure 12: Qualitative results from *sketch → anime* task.

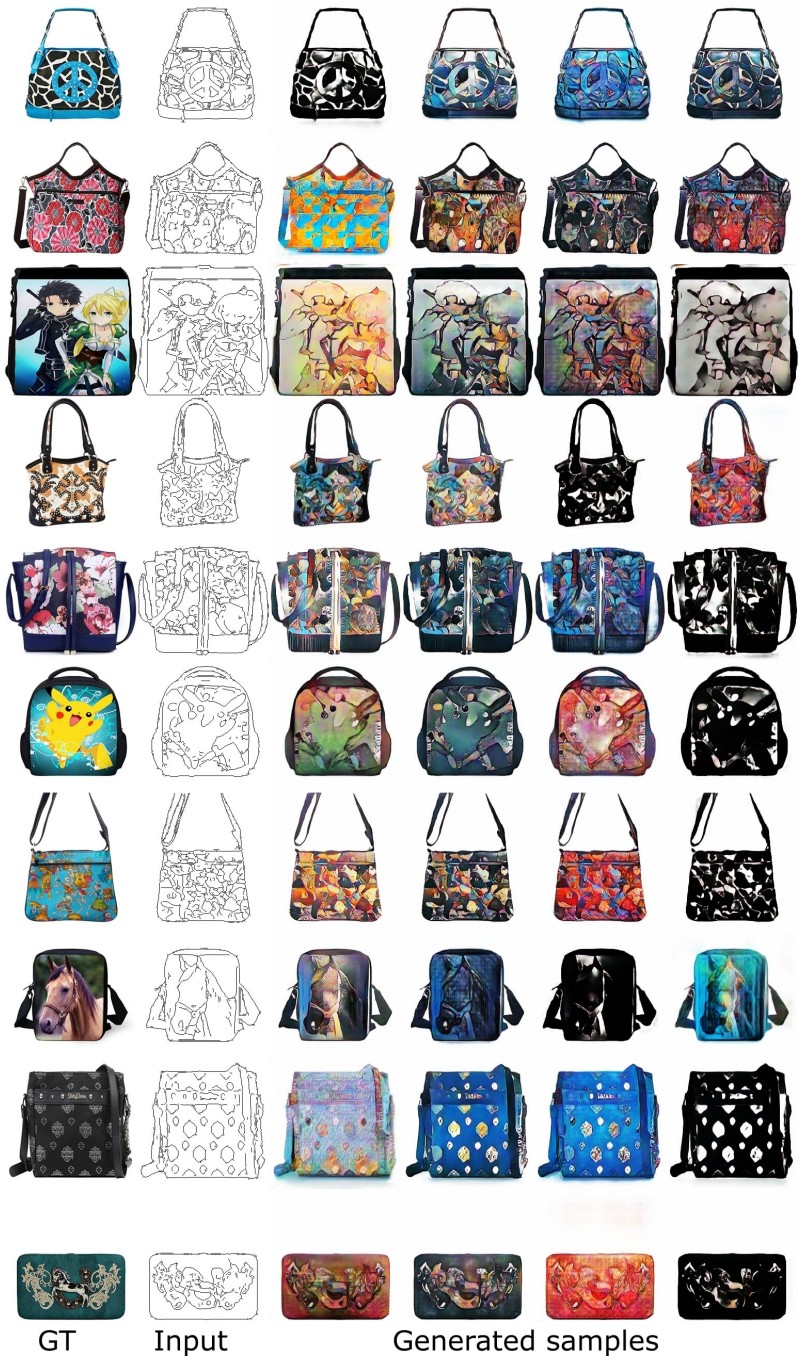

GT     Input         Generated samples

Figure 13: Qualitative results from *sketch → bags* task.

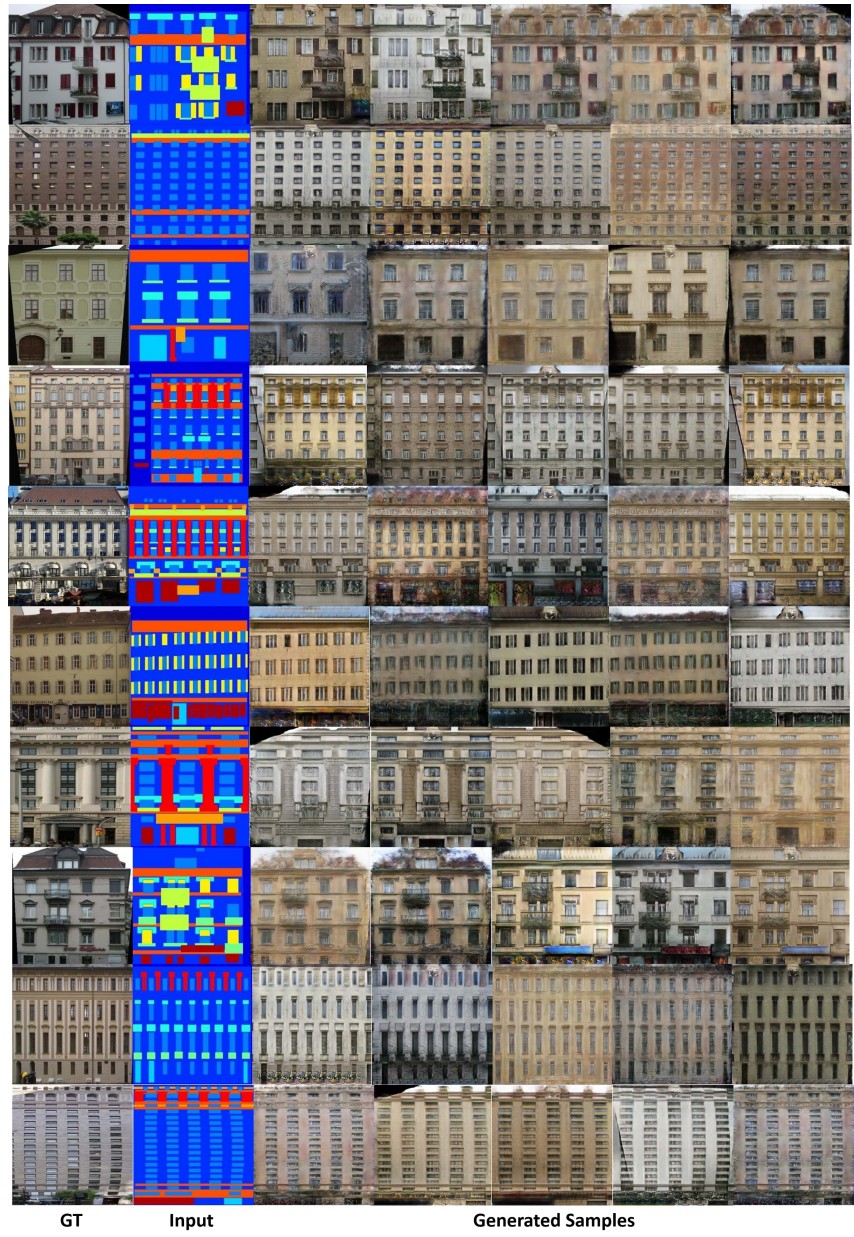

GT        Input                    Generated Samples

Figure 14: Qualitative results from *labels → facades* task.

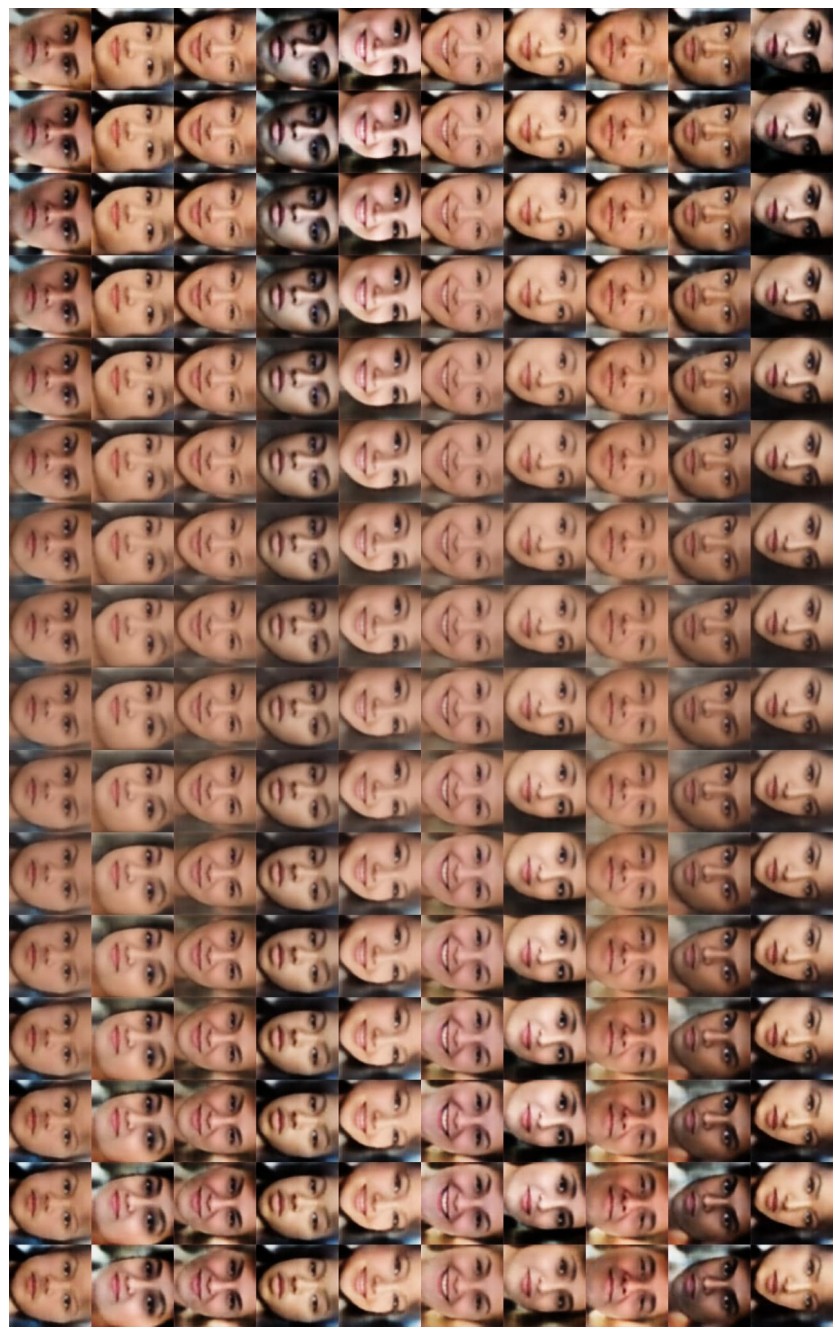

Figure 15: Smooth interpolations of our model. Each column represents an interpolation between two latent codes, conditioned on an input. The faces are rotated to fit the space.