# OpenReview forum: "Rethinking conditional GAN training: An approach using geometrically structured latent manifolds"
_NeurIPS.cc/2021/Conference — NeurIPS 2021 Poster_

### Official Review · Reviewer_X71S · 2021-07-05

**Rating:** 7
**Confidence:** 4

**Summary:**

This paper attempts to address poor sample quality and sample diversity in image conditioned generative models by addressing the topology mismatch between latent space and image space. It introduces two new loss functions: one which encourages Euclidean paths in the latent space to map to geodesic paths in image space, and another which helps to enforce a bijective mapping between the latent manifold and the output image manifold. Experimental results demonstrate quantitative and qualitative improvement over previous methods.

**Limitations And Societal Impact:**

Paper does not discuss negative societal impact of work.

**Main Review:**

Strengths:
 - Good theoretical motivation for the proposed loss functions.
 - Well written and fairly easy to follow.
 - I enjoyed the analysis of model properties such as colour distribution and interpolation behaviour. Figure 4 in particular does a great job of demonstrating why we might want good mappings between Euclidean latent paths to geodesics in image space.

Weaknesses:
- The main contribution of matching Euclidean paths in the latent manifold to geodesic paths in the image manifold seems to have been proposed before as "path length regularization" in StyleGAN2 [1]. Can the authors comment on the differences between path length regularization and their method?
- Performance improvement over existing methods is not clear. Proposed methods achieves better image quality on only 4 of 9 datasets, and better diversity on 6 of 9 datasets. That being said, performance improvement over the baseline is significant. Could the proposed method be combined with other SOTA methods to achieve better performance across all datasets?


Questions:
- The loss in Equation 10 appears to require several iterations for each training batch. How does this affect training time compared to the baseline model?
- Random perturbations such as brightness and contrast are applied to real images in Equations 12 and 13 in order to improve sample diversity. How does this compare to simply applying these augmentations to the full training set?


Other comments:
- Line 27: May want to explain more about the significance of dropout in case readers are not aware of its function in Pix2Pix.
- Line 157: Potential typo: "...is a continues global chart map". Should "continues" be "continuous"?
- Table 1: It seems weird to me to use the incomplete acronym LPIP (Learned Perceptual Image Patch). Using the full acronym LPIPS (Learned Perceptual Image Patch Similarity) would make more sense.
- Figure 1 does not include comparison to CGML despite it being one of the better models being compared against. It may be good to include some comparison with CGML as well so as to not mistakenly mislead readers about the quality improvements compared to previous methods.

Overall, I thought this was a good paper. My main concern is about how similar the proposed method is to path length regularization. If the authors could demonstrate that their method outperforms path length regularization I would feel much better about accepting this paper.

[1] Karras, Tero, et al. "Analyzing and improving the image quality of stylegan." Proceedings of the IEEE/CVF Conference on Computer Vision and Pattern Recognition. 2020.

---
After rebuttal:
My concerns have been addressed. I have increased my score from 5 to 7.

**Time Spent Reviewing:**

6

---

> ### Author Response · Authors · 2021-08-10
> **Response to R4**
>
> We appreciate the insightful comments by the reviewer. Please see our answers below:
>
> **1. The main contribution of matching Euclidean paths in the latent manifold to geodesic paths in the image manifold seems to have been proposed before as "path length regularization" in StyleGAN2. Can the authors comment on the differences between path length regularization and their method?**
>
> We thank the reviewer for bringing this paper to our attention. We point out two main differences between ours and path length regularization approach:
>
> 1. The path length regularization objective proposed in [a] is minimized when the Jacobian of the generator function is orthogonal. This forces the path lengths of the latent space to be *equal* to the paths on the data manifold (up to a global scale). More precisely, let $\gamma(t):\mathbb{R} \to M_z$ is a curve defined on the latent manifold. Then, $\int_{t_0}^{t_1}|\gamma'(t)|dt = \int_{t_0}^{t_1}| (G \circ \gamma)'(t)|dt$. In other words, $G$ becomes a (metric) isometry, which can be *undesirable* because it restricts the class of functions that $G$ can approximate. It can be shown that if a manifold is isometric to a Eucledian space (such as in the above case), then the Riemannian curvature tensor on the manifold vanishes identically at each point. It is clear that such a representation has limitations towards approximating a complex image manifold. In contrast, we only encourage the shortest paths on the latent manifolds to map to geodesics on the output manifold, where the arc lengths of the geodesics can vary significantly (within a bounded interval) depending on the shape of the manifold along each direction. This allows our generator to produce complex output manifolds without being restricted to isometries while still allowing smooth interpolations.
>
> From another angle, the path length regularization object encourages a global condition where $\frac{\partial G}{\partial z}$ across every direction to be constant. This hampers the ability of the generator model complex manifolds. In contrast, we only encourage minimization of $\frac{\partial G}{\partial z}$ along curves in the latent space, as much as possible. The magnitude of $\frac{\partial G}{\partial z}$ at each point (hence geodesic lengths), can vary depending on the initial velocity of the curve in each iteration, ground truth, and the choice of $\alpha(\cdot)$.
>
>
>  2. A direct consequence of such a restrictive constraint can be a contradiction against other loss components of the model. This is also stated in the original paper [a] as *The full loss function of the training is a combination of potentially conflicting criteria, and it is not clear if a genuinely isometric mapping would be capable of expressing the image manifold of interest*. In contrast, we did not observe such a drawback with our loss functions.
>
> To empirically verify the above claims, we conducted an experiment, where we replaced $L_{gh}$ with the path length regularization objective $L_{pl}$, on Pix2Pix, against the landmarks-to-faces task. Through exhaustive weight tuning of the loss function, we were able to obtain a best score of $(LPIPS, FID) = (0.109, 81.7)$ which is inferior to $L_{gh}$ where we obtained $(LPIPS, FID) = (0.197, 67.82)$. However, please note that there is a trade-off. When the $L_{pl}$ has a higher weight, it leads to lower image quality, but higher diversity. On the other hand, if $L_{pl}$ has a lower weight, the other loss functions absorb the effect of it, indicating the loss component contradiction we mentioned under point two. Empirically, we observed an (approximately) concave function for the image quality and a monotonically increasing function for the diversity, against $L_{pl}$, when other conditions are freezed.
>
> We will cite the paper and include these results in the ablation study in Table 2.
>
> **2. Performance improvement over the baseline is significant. Could the proposed method be combined with other SOTA methods to achieve better performance across all datasets?**
>
> Thank you for the suggestion. Please note that the main goal of the paper was to propose a geometrically inspired generic training mechanism that can improve the performance of classic generative models significantly, without explicit modifications to the architecture. For instance, we apply our algorithm to four different networks that specialize in different tasks and achieve improved performance.
>
> Nevertheless, since the concept behind our algorithm is architecture agnostic, it can be easily injected into any suitable model. For instance, our latest experiments showcase that with our algorithm, we can improve the diversity and the structure of the latent space of StarGANv2, without any modifications to the architecture.  As a result, the diversity of the StarGANv2 is increased, along with better interpolation characteristics along the style space. For example, on CelebA-HQ, we were able improve $(FID|LPIPS):(13.7|0.452)\to(13.1|0.488)$. To this end, we simply replace the $L_{ds}$ of StarGANv2 with our loss $L_{gh}$. In StarGANv2, the images  are generated as a composition of two functions: the style generator ($F(\cdot)$) and the image generator ($G(\cdot, \cdot)$). We enforce the bi-lipschitz constraint between the latent space and the image generator output. It can be easily shown that for continuously differentiable functions $F$ and $G$, if the composition $G \circ F$ is homeomorphic, $F$ and $G$ are individually homeomorphic. Hence, the image generator output becomes homeomorphic to the style space.
>
> We will include detailed results with more experiments in the Appendix. Importantly, note that since our training algorithm is applied to an existing model, the quality of the outputs is somewhat bound to the performance of the original model (although our algorithm increases the quality). For instance, when applied to a baseline model like Pix2Pix, although the quality of the outputs is increased significantly, it may not be able to directly compete with a generative model such as StarGANv2 (in terms of the realism of outputs), where various explicit network components act in unison to generate high-quality outputs. In contrast, when our method is applied to StarGANv2, we observe high-quality results (with more diversity and structure), since the baseline model is already powerful in this aspect.
>
> **3. The loss in Equation 10 appears to require several iterations for each training batch. How does this affect training time compared to the baseline model?**
>
> With a single GTX 1080Ti GPU, the training time for the Pix2Pix-Geo is $5.8$x higher than baseline Pix2Pix. However, we observed that the stand-alone training times can be significantly improved with multiple GPUs. Note that number of trainable parameters and inference time are same for both models.
>
> **4. Random perturbations such as brightness and contrast are applied to real images in Equations 12 and 13 in order to improve sample diversity. How does this compare to simply applying these augmentations to the full training set?**
>
> Thank you for the suggestion. We have already tried this experiment and would like to confirm that without explicitly enforcing a bi-lipzchitz mapping between the manifolds, the generator simply cannot improve the diversity. We will mention this in the  appendix of the revised paper.
>
> **5. Line 157: Potential typo: "...is a continues global chart map". Should "continues" be "continuous"?**
>
> Thank you for pointing out our mistake. We will correct it in the revision.
>
> **6. Table 1: It seems weird to me to use the incomplete acronym LPIP (Learned Perceptual Image Patch). Using the full acronym LPIPS (Learned Perceptual Image Patch Similarity) would make more sense.**
>
> Thank you. We will fix this.
>
> **7. May want to explain more about the significance of dropout in case readers are not aware of its function in Pix2Pix.**
>
> Thank you for the suggestion. We will include this in revision.
>
> **8. It may be good to include some comparison with CGML as well so as to not mistakenly mislead readers about the quality improvements compared to previous methods.**
>
> We are grateful for the suggestion. We will add more qualitative experiments against CGML in the Appendix.
>
> **References**
>
> [a] - Karras, Tero, et al. ``Analyzing and improving the image quality of stylegan." Proceedings of the IEEE/CVF Conference on Computer Vision and Pattern Recognition. 2020.

---

> > ### Comment · Reviewer_X71S · 2021-08-19
> > **Post-rebuttal Thoughts**
> >
> > Thank you authors for the detailed and complete response! My concerns and questions have been addressed and I will increase my score accordingly.

---

> > > ### Author Response · Authors · 2021-08-22
> > > **Thank you**
> > >
> > > We thank the reviewer for raising the score. We gratefully acknowledge that the questions and suggestions from the reviewer helped us to improve our current manuscript.

---

### Official Review · Reviewer_9CLc · 2021-07-16

**Rating:** 7
**Confidence:** 3

**Summary:**

This paper suggests that the current way of training conditional generative models can be the root of the mode-collapsing problems and proposes a fix at a fundamental level. By imposing bi-Lipschitz constraint to the relationship between latent space and image space, the baseline model shows largely improved diversity and quality without any architectural modifications.

**Ethical Concerns:**

no comments

**Limitations And Societal Impact:**

no comments

**Main Review:**

This is a very nice work with impressive analyses and the paper is very well written.


In particular, I am very impressed with how systematically the authors investigate the deficiencies of the current state-of-the-art. The way that the authors analyze the problem is very convincing, and the theoretical analyses provide a good insight on the prevailing problem in conditional generative models at a fundamental level. I believe this insight will be useful for the future developments of the field.


My main concern is on the quality of the results and the baseline models that the authors used. Although Pix2Pix is a good baseline to showcase the prominent problem that is being addressed, the results would have looked more persuasive if they were compared with more recent image-to-image translation models that addressed the intra/inter domains diversity such as, MUNIT, StarGANv2 , etc. (since the method can be applied generally.) The quantitative and qualitative results largely fall behind to those of the current state-of-the-arts, which keeps me from giving a strong score to this paper. (since the practical impact is also a very important aspect in this field.)


Related to the previous comment, I am curious if this would be possible to be applied to more aggressive translations such as among different domains? (Since it is constraining the relationship quite strongly, I am not sure if this can be applicable to add any new features that exist in the new domain, e.g., to translate a dog to a lion, one needs to draw its mane.)


Line 121 the distortion of M_z with respect to M_z. -> I guess one of these two must be M_y?


Line 162 _I_ is not defined. I guess, this stands for an interval?

---

After rebuttal (update): I raise the score to 7.

---


**Time Spent Reviewing:**

7

---

> ### Author Response · Authors · 2021-08-10
> **Response to R3**
>
> We thank the reviewer for insightful comments. Please see our answers below:
>
> **1. Although Pix2Pix is a good baseline to showcase the prominent problem that is being addressed, the results would have looked more persuasive if they were compared with more recent image-to-image translation models that addressed the intra/inter domains diversity such as, MUNIT, StarGANv2, etc. (since the method can be applied generally.)**
>
> Thank you for the suggestion. Please note that the main goal of the paper was to propose a geometrically inspired generic training mechanism that can improve the performance of classic generative models significantly, without explicit modifications to the architecture. For instance, we apply our algorithm to four different networks that specialize in different tasks and achieve improved performance.
>
> Nevertheless, since the concept behind our algorithm is architecture agnostic, it can be easily injected into any suitable model. For instance, our latest experiments showcase that with our algorithm, we can improve the diversity and the structure of the latent space of StarGANv2, without any modifications to the architecture.  As a result, the diversity of the StarGANv2 is increased, along with better interpolation characteristics along the style space. For example, on CelebA-HQ, we were able improve $(FID|LPIPS):(13.7|0.452)\to(13.1|0.488)$. To this end, we simply replace the $L_{ds}$ of StarGANv2 with our loss $L_{gh}$. In StarGANv2, the images  are generated as a composition of two functions: the style generator ($F(\cdot)$) and the image generator ($G(\cdot, \cdot)$). We enforce the bi-lipschitz constraint between the latent space and the image generator output. It can be easily shown that for continuously differentiable functions $F$ and $G$, if the composition $G \circ F$ is homeomorphic, $F$ and $G$ are individually homeomorphic. Hence, the image generator output becomes homeomorphic to the style space.
>
> We will include detailed results with more experiments in the Appendix. Importantly, note that since our training algorithm is applied to an existing model, the quality of the outputs is somewhat bound to the performance of the original model (although our algorithm increases the quality). For instance, when applied to a baseline model like Pix2Pix, although the quality of the outputs is increased significantly, it may not be able to directly compete with a generative model such as StarGANv2 (in terms of the realism of outputs), where various explicit network components act in unison to generate high-quality outputs. In contrast, when our method is applied to StarGANv2, we observe high-quality results (with more diversity and structure), since the baseline model is already powerful in this aspect.
>
> **2. Line 121 the distortion of $M_z$ with respect to $M_z$.  I guess one of these two must be $M_y$?**
>
>  Thank you for pointing out our mistake. We will fix this in our revision.
>
> **3. Line 162 I is not defined. I guess, this stands for an interval?**
>
> Yes, $I$ is an interval in $\mathbb{R}$. Please note that we have mentioned this in L163.

---

> > ### Comment · Reviewer_9CLc · 2021-08-19
> > **Thank you for your response.**
> >
> > The authors have addressed my concerns, and I raise my score. One suggestion: Now that the algorithm is shown effective when combined with the current SOTA in image translations, it would be nice to include these results in the main manuscript and change the results accordingly (I know this is quite a demanding job, so no pressure, but I suggest this since I think that this paper can look better with those.)

---

> > > ### Author Response · Authors · 2021-08-22
> > > **Thank you**
> > >
> > > We appreciate the positive feedback from the reviewer and are grateful for raising the score. We will try our best to include experiments over all the datasets using a SOTA GAN in the final revision.

---

### Official Review · Reviewer_agx2 · 2021-07-19

**Rating:** 7
**Confidence:** 3

**Summary:**

This paper tries to improve the current limitation of conditional GANs with a novel topological space matching view. The authors claim to solve the main drawbacks of existing cGANs that cause the GAN to lack diversity and randomness. Basically, they propose an optimization algorithm to encourage the generator and the latent space to preserve a bi-lipschitz mapping between generated space and GT space. The authors compare evaluate their method on image translation task and show higher quality of generator comparing with existing SOTA cGANs.

**Limitations And Societal Impact:**

This paper potentially has impacts.

**Main Review:**

+ This paper theoretically analysis the main drawback of existing cGANs in view of space matching. It is very interesting to explain the current drawback of GANs (mode collapse, insensitivity to randomness) using Riemannian geometry.
+ The authors provide detailed proof of their claim and theory.
+ The experimental results show significant improvement on image translation tasks comparing with existing major cGANs,.

This is a good paper. There are still some flaws in this paper.
	1. This paper lacks essential visualization of helping to understand. Some parts of their explanation are not easy to follow.
	2. Some notations in section 3.1 are not cleared. It is suggested to summarize all notations in a list before going into the math part.
	3. It is encouraged to evaluate the method on wider datasets and tasks. For example, evaluate it on sentence generation or cross-domain generation tasks.


**Time Spent Reviewing:**

4

---

> ### Author Response · Authors · 2021-08-10
> **Response to R2**
>
> We are grateful for the encouraging and valuable comments. Please see our answers below.
>
> **1. This paper lacks essential visualization of helping to understand.**
>
> Thank you for the suggestion. We will add a small figure to the introduction to better explain our method. Specifically, we will illustrate that our approach induces a bilipschitz mapping between the latent and output manifolds, while mapping geodesics to geodesics. The process of encouraging global and local homeomorphisms will also be indicated in the same figure.
>
> **2. Some notations in section 3.1 are not cleared. It is suggested to summarize all notations in a list before going into the math part.**
>
> Thank you for the suggestion. We will add a short paragraph at the beginning of the method section to clarify the math notations.
>
> **3. It is encouraged to evaluate the method on wider datasets and tasks. For example, evaluate it on sentence generation or cross-domain generation tasks.**
>
> We will add cross domain generative experiments to the Appendix based on StarGANv2. By using our algorithm, we were able to increase the diversity of the StarGANv2, along with better interpolation characteristics along the style space. For example, on CelebA-HQ, we were able improve $(FID|LPIPS):(13.7|0.452)\to(13.1|0.488)$. Please refer our answer to reviewer 3 for a detailed explanation.

---

### Official Review · Reviewer_q3Au · 2021-07-22

**Rating:** 6
**Confidence:** 4

**Summary:**

To tackle the problem of mode collapse in conditional generative adversarial networks (C-GAN), the paper introduces a new loss function and an associated training procedure, inspired by differential geometric considerations. In particular, the mapping learnt by the generator network is required to be bi-Lipschitz so that it preserves the local Euclidean geometry between the latent and data spaces (the tangent space of the manifold) at every point. The proposed approach is empirically validated on a variety of image to image translation tasks.

**Limitations And Societal Impact:**

Yes.

**Main Review:**

#1. Originality/ Novelty: The approach builds upon the work in [18] by analyzing the issue of manifold distortion to propose a novel loss-function to encourage local bijectivity and global bi-Lipschitz mapping to simultaneously address the problems of mode collapse and latent space distortion. The proposed methodology is novel to my knowledge.

#2. Prior Art: The prior art is adequately cited.

#3. Technical Soundness: The technical formulation appears okay subject to the following queries which the authors should address in the response or fix in the paper. I did check the math in the paper and the Appendix but not completely and rigorously.

Q1. (line 17-20) Shouldn't the reconstruction loss be between y and G(x,z)?

Q2: (line 31-35) It's not clear what is meant here:  Aren't the latent space variables the chart variables for the manifold in the data space and for a bijective onto map, locally the map is well approximated as an invertible affine transformation,  i.e. locally, the z-space and the tangent space look very similar to each other and shortest distances will map to each other ? Conversely, if 'all' shortest distances on the data manifold correspond to shortest distance between the pre-images in the latent space, then the mapping function has to be linear (affine)?

Q3. (lines 36-38) "naïve coupling of regression and adversarial loss..." This should be carefully framed. Objectives having multiple terms mostly lead to a compromise.

Q4: (lines 45-51) Shouldn't bijective homeomorphism imply that dim(Y) = dim(X) + dim(Z)? In other words, the dimensionality of the latent space, by design, should exactly equal the different between the intrinsic dimensionality of the output and input spaces? If so, how is this ensured? Also, since a locally bijective homeomorphic will preserve shortest distances (locally), is the goal to have this property preserved somewhat non-locally (it can't be globally ensured)?

Q5: Section 2.1, Eqn. (3) and Proof in Appendix D: \bar{y}(z)  is a function (z). It seems like it is being treated erroneously as independent of z and therefore \bar{y}(z) appears in the limits of the outer integral with the integrating variable y while the inner integrating variable is z. This doesn't seem correct.

Q6: Section 2.3 lines 112-120: Generators with a continuous activation function are compositions of continuous functions and hence continuous. Furthermore, bijectivity is enforced by the Bicycle-GANs, and hence Bicycle-GANs should be homeomorphic as well. However, it is correct that the local gradients and the Lipschitz constant of the G network are not explicitly bounded.

Q7: Since locally bi-Lipschitz mapping would suffice to reduce mode collapse as well as bound the distortion, why does a simpler loss function with an additional term similar to (but in the other direction) Equation (2) in [18] doesn't suffice and the mechanism in 3.1 is needed? Even if needed, shouldn't the simpler approach be tried as a baseline?

#4. Technical details and reproducibility: Reasonable technical details are provided in the paper. The methodology is fairly involved and may not be reproducible by itself. However, the authors provide the code (which I have not tested) which should help reproduce the results.

#5. Experimental Evaluation: Results on a variety of tasks are provided. They seem promising though not compelling. For example, the quantitative results in Table 1 are promising (better in majority of cases) but not conclusive with respect to MR-GAN [4], DS-GAN [18] and Bicycle-GAN [20]. Images are too small for qualitative evaluations. Interpolations and the ablation study bolsters the presented approach.

#6. Clarity: The paper needs to be written more carefully (see my queries in point #3 above).

- It is stated in line 160 that R1 implicitly resolves the loss mismatch, while there are details with regards to this in the supplementary material, an intuitive explanation of the same in the main paper would be helpful given that it is an important motivation for the proposed method.

- In section-2.2, since the definition and proposition are exactly the same as presented in [18], it might be more useful to present an intuitive explanation of the relevant concepts here, rather than reproducing the definition/proposition verbatim.

Minor typos:
-- Capital letters should be used for acronyms in the references.
-- line 14: 'learns' --> 'learn'
-- line 121: M_y instead of M_z

#7. Significance:  Using insights from differential geometry to analyze the mode collapse and the manifold distortion problem is good addition to this line of work. The approach is reasonable and the experimental results promising. As the community verifies and reproduces the results, the paper has a potential to be a significant addition to the field.

#8. Preliminary rating and justification: A good paper addressing an important problem in the space of generative models with potentially wide scope for impact. Most ideas presented are well motivated; the proposed method seems novel and is well supported by relevant experimental results. Though there is a need to improve clarity and address my concerns and queries about technical soundness in #3. Hence, I recommend the acceptance of the paper subject to the author response and other reviews.

**Time Spent Reviewing:**

10

---

> ### Author Response · Authors · 2021-08-10
> **Response to R1**
>
> We thank the reviewer for insightful comments. Please see our answers below.
>
> **1. (line 17-20) Shouldn't the reconstruction loss be between $y$ and $G(x,z)$?**
>
> Yes, the reconstruction loss is applied between $y$ and $G(x,z)$. Since $y$ depends on $x$, the goal of the reconstruction loss is to force the generator to produce outputs that strongly depend on $x$. When the adversarial loss $L_{adv}$ is used without the reconstruction loss $L_r$, the generator tends to forget the conditioning ($x$) although the discriminator sees both $x$ and $y$. This is a common observation in the literature (see Fig. 4 in [a]; [b]). The above phenomenon is the key reason that $L_r$ is used with $L_{adv}$ in previous works, i.e., to strongly condition the generator on the input. In fact, this conditioning has to be so strong, the weight of $L_r$ is around $100$ times higher than $L_{adv}$ (e.g., Pix2Pix). We will clarify this in the revised paper.
>
> **2. (line 31-35) It's not clear what is meant here: Aren't the latent space variables the chart variables for the manifold in the data space?**
>
> Not necessarily. For the latent variables to be the chart variables of the output manifold, a homeomorphism should exist between the above spaces (by definition). If explicit priors are not enforced, the homeomorphism is violated due to mode collapse (e.g., Pix2Pix). Even with priors, homeomorphism cannot be guaranteed. This is indicated in Fig.~4, that is, large areas of the latent space point to a single point on the output manifold for the case of DSGAN. Hence, the inverse of $G$ does not exist, i.e., the mapping is not a homeomorphism. In contrast, ensuring the existence of this continuous inverse is precisely what we are trying to enforce via R1 (in L155-157).
>
> **3. For a bijective onto map, locally the map is well approximated as an invertible affine transformation, i.e. locally, the z-space and the tangent space look very similar to each other and shortest distances will map to each other ?**
>
> We would like to clarify that a bijective map from the latent space to the output manifold *cannot* be simply approximated as an affine transformation in most cases. Any affine transformation $f:U \to V$ can be written as $f(\cdot) = A(\cdot) + v$, where $v \in V$ and $A(\cdot)$ is a linear transformation. In contrast, note that a bijective onto map between two manifolds covers a highly complex class of functions. See a simple example below:
>
> Let $G(x) = [\frac{e^{(x-\mu_1)^2}}{\sigma},\frac{e^{(x-\mu_2)^2}}{\sigma}, \dots \frac{e^{(x-\mu_n})^2}{\sigma}]$. One can check that $G$ induces a local homeomorphism (hence a bijective onto map) between $\mathbb{R}$ and a 1-D manifold embedded in $\mathbb{R}^n$. However, it is not affine.
>
> Further, we do not enforce the geodesics arc lengths between the manifolds to be exactly equal to each other. If they are equal, that means we have an isometry between the spaces, hence (possibly) an affine transformation. In contrast, we only map geodesics to geodesics (where arc lengths of the geodesics can differ) which allows better structure between the manifolds. In an unrestricted setting, this would not happen automatically. A clear visualization is shown in Fig. 1 in [c].  Observe that in order to move along the geodesics in the output manifold, the latent variables have to move along non-geodesics. This behavior is more extreme with conditional GANs due to conditional mode collapse.
>
> **4. Conversely, if 'all' shortest distances on the data manifold correspond to the shortest distance between the pre-images in the latent space, then the mapping function has to be linear (affine)?**
>
>  Not necessarily. Gnomonic projection [d] is a simple counter-example. It induces a homeomorphic map between $\mathbb{S}^n$ and $\mathbb{R}^n$ while mapping the geodesics on $\mathbb{S}^n$ to shortest paths in $\mathbb{R}^n$. Yet, it is not an affine transformation.
>
> **5. (lines 36-38) "naïve coupling of regression and adversarial loss..." This should be carefully framed.**
>
>  Thank you for pointing this out. We will rephrase this statement appropriately.
>
> **6. (lines 45-51) Shouldn't bijective homeomorphism imply that dim(Y) = dim(X) + dim(Z)? In other words, the dimensionality of the latent space, by design, should exactly equal the difference between the intrinsic dimensionality of the output and input spaces? If so, how is this ensured?**
>
> A more accurate statement would be $dim(y) = dim(x \oplus z)$, where $\oplus$ is concatenation (since the dimension of a concatenation of subspaces does not necessarily equal to the addition of individual subspace dimensions). Importantly, note that we have to consider the dimensionality of the manifolds *themselves* and not the dimensionality of the ambient spaces of the manifolds. For instance, the dimensionality of the ambient space of the output manifold with $32 \times 32$ images is $\mathbb{R}^{1024}$. Now consider the generator $G(x,z)$. As soon as we ensure a homeomorphism between $x \oplus z$ and $G(x,z)$, the true dimensionality of the output manifold automatically becomes $dim(x \oplus z)$, regardless of the ambient space. No other explicit constraints are needed. As a simple example, consider $G(x,z) = [\frac{e^{(x-\mu_1)^2}}{\sigma},\frac{e^{(x-\mu_2)^2}}{\sigma}, \dots \frac{e^{(x-\mu_n})^2}{\sigma}, \frac{e^{(z-\mu_1)^2}}{\sigma},\frac{e^{(z-\mu_2)^2}}{\sigma}, \dots \frac{e^{(z-\mu_n})^2}{\sigma}]$, where $x,z \in \mathbb{R}$. One can check that a homemorphism exists between $\mathbb{R}^2$ and the output manifold of the $G(x,z)$ (recall that a continuous bijection from a compact space to a Hausdorff space is a homeomorphism). Therefore, although the codomain of $G(x,z)$ lives in $\mathbb{R}^{2n}$ ambient space, it is actually a 2D manifold.
>
> In fact, this (manifold hypothesis) is the fundamental motivation behind all latent variable generative models; although image manifolds reside in a higher dimensional ambient space, the actual dimensionality of the manifold is extremely less. That is why we can represent the generated manifold using a low-dimensional latent space.
>
> **7. Also, since a locally bijective homeomorphic will preserve shortest distances (locally), is the goal to have this property preserved somewhat non-locally?**
>
> We would like to clarify that locally bijective homeomorphisms will not necessarily preserve shortest distances locally. Shortest distance preservation is a much more restrictive condition. See a counter example below:
>
> Consider $\mathcal{M} = \mathbb{S}^2 - \{ (0,0,1), (0,0,-1) \}$, i.e., the unit sphere without the north and south poles. Again consider $\mathcal{N} = [-\pi, \pi] \times [-\pi/2, \pi/2]$. Now, let $f:\mathcal{M} \to \mathcal{N}$ be the *plate caree* projection [e], i.e., $f$ maps from the sphere to a plane (without north and south poles). Now, this is a homeomorphism. However, note that the arcs along the great circles on the sphere do not *necessarily* map to straight lines on the plane (recall that the geodesics on the sphere are arcs along the great circles).
>
>  This is the reason we encourage shortest paths in the latent space map to minimum distortion paths in the output manifold. Since geodesics avoid large distortions, this encourages geodesic mappings between the manifolds.
>
> **8. Since locally bi-Lipschitz mapping would suffice to reduce mode collapse as well as bound the distortion, why does a simpler loss function with an additional term similar to (but in the other direction) Equation (2) in [18] doesn't suffice and the mechanism in 3.1 is needed?**
>
> Thank you for the suggestion. However, we attempted this method in our preliminary experiments without success. Consider the minmization objective below:
>
> \begin{equation}
>     \beta_1 L_{adv} - \beta_2\mathrm{min}( \frac{||G(x,z_1) - G(x,z_2)||}{||z_1 - z_2||} ,\tau_1) - \beta_3 \mathrm{min}( \frac{||z_1 - z_2||}{||G(x,z_1) - G(x,z_2)||}, \tau_2)
> \end{equation}
>
> In practice, the second loss term does not affect the whole latent spaces systematically. Observe the interpolation results of Fig.~4 in our paper. It is clear that there are large intervals in the latent space, which point to extremely small intervals in the output manifold.  In these areas, the third loss term will always be equal to the constant $\tau_2$ and will never be optimized. In summary, adding the third loss term would not have an effect on those areas, and the bi-lipschitz conditions will not be enforced.
>
> On the other hand, removing $\tau_2$ or $\tau_1$ causes extremely large gradients and the model convergence cannot be achieved in practice.
>
> We will add this discussion in the Appendix.
>
> **References**
>
> [a] - Isola, Phillip, et al. "Image-to-image translation with conditional adversarial networks." Proceedings of the IEEE conference on computer vision and pattern recognition. 2017.
>
> [b] - Lee, Soochan, Junsoo Ha, and Gunhee Kim. "Harmonizing maximum likelihood with GANs for multimodal conditional generation." arXiv preprint arXiv:1902.09225 (2019).
>
> [c] - Arvanitidis, Georgios, Lars Kai Hansen, and Søren Hauberg. "Latent space oddity: on the curvature of deep generative models." arXiv preprint arXiv:1710.11379 (2017)
>
> [d] - https://en.wikipedia.org/wiki/Gnomonic_projection
>
> [e] - https://en.wikipedia.org/wiki/Equirectangular_projection
>
> [f] - Ramasinghe, Sameera, et al. "Conditional Generative Modeling via Learning the Latent Space." arXiv preprint arXiv:2010.03132 (2020).
>
> [g] - https://soochanlee.com/publications/mr-gan.html
>
> [h] - Geirhos, Robert, et al. "Shortcut learning in deep neural networks." Nature Machine Intelligence 2.11 (2020): 665-673.

---

> ### Author Response · Authors · 2021-08-10
> **Response to R1 continued..**
>
> **9. Section 2.1, Eqn. (3) and Proof in Appendix D: $\bar{y}(z)$ is a function ($z$). It seems like it is being treated erroneously as independent of z and therefore $\bar{y}(z)$ appears in the limits of the outer integral with the integrating variable y while the inner integrating variable is $z$. This doesn't seem correct.**
>
> We apologize for the incorrect notation. The derivation should be corrected as follows:
>
> Let $y^*$ be any estimate for $y$. Then,
>
> \begin{equation}
>      E_{y,z}|y - y^*|= \int_{\infty}^{\infty} \int_{\infty}^{\infty} |y - y^*|p(y)p(z|y)dzdy.
> \end{equation}
>
> Skipping intermediate steps, at the minimization of the above,
>
> \begin{equation}
>         \int_{\infty}^{y^*}p(y)dy = \int_{y^*}^{\infty}p(y)dy.
> \end{equation}
>
> Note that the above derivation implicitly assumes that the estimate $y^*$ is independent of $z$. In other words, the above derivation shows that there exists a perfect solution for the generator which satisfies the reconstruction loss, if it enforces $y^*$ to be independent of $z$.
>
> Now, this raises to the interesting question: *is there another solution that satisfies the reconstruction loss and the adversarial loss simultaneously?* In fact, theoretically, there is. If the generator can make sure that $p(y) \equiv p(y^*)$ by taking $z$ into account, the adversarial and reconstruction losses will both be satisfied. But unfortunately, this does not happen in practice, *i.e.*, the generators almost always go for the easier solution (making $y^*$ independent of $z$) under the traditional loss. This is the reason that networks such as BicycleGAN, DSGAN, MRGAN, and CGML etc., have to opt for alternative approaches. We provide two insights regarding this aspect below:
>
> 1. This phenomenon ($y^*$ becoming independent of $z$ under the reconstruction loss) has been empirically observed in the literature (see Fig. 3 in [f] and Fig. 3 from the top in [g]). Further, from an alternative theoretical perspective, [b] showed that $z$ indeed tends to be independent of $y$, i.e., the generator forgets $z$, and learns to predict the conditional mean for $\ell_1$ loss.
>
> 2. By making $p(z|y^*) = p(z)$ the $G$ can easily find a set of parameters that can minimize $\ell_1$ loss, i.e., there exists a short cut solution for the network, if it forces $p(z|y^*) = p(z)$. It is widely observed that deep networks always tend to go for the lazy solution [h], which happens in this scenario as well. While being an open research problem, a possible reason for this might be that deep networks implicitly follow *minimum description length* principle: $T_{best} = argmin_{T} [L(T) + L(D|T)]$, where $L(T)$ is the description length  of the model $T$ in bits, and $L(D|T)$ is the description length  of the data $D$ in bits when encoded with $T$. Note that $L(D|T)$ decreases when $p(z|y^*) = p(z)$ since the model does not have to take $z$ into account. However, please note that finding the exact root cause for this phenomenon is out of scope for this work.
>
> We thank the reviewer for raising this interesting question and we will include this discussion in the Appendix.
>
> **10. It is stated in line 160 that R1 implicitly resolves the loss mismatch, while there are details with regards to this in the supplementary material, an intuitive explanation of the same in the main paper would be helpful given that it is an important motivation for the proposed method.**
>
> Thank you for pointing this out. We will add a short explanation to the methodology section.
>
> **11. In section-2.2, since the definition and proposition are exactly the same as presented in [18], it might be more useful to present an intuitive explanation of the relevant concepts here, rather than reproducing the definition/proposition verbatim.**
>
> Thank you for the suggestion. We believe that a rigorous statement of the proposition would be useful to the discussion. However, following your suggestion, we will add an intuitive explanation regarding the proposition and the definition.
>
> **12. Typos**
>
> Thank you for pointing out our mistakes. We will correct them in the revision.
>
> **References**
>
> [a] - Isola, Phillip, et al. "Image-to-image translation with conditional adversarial networks." Proceedings of the IEEE conference on computer vision and pattern recognition. 2017.
>
> [b] - Lee, Soochan, Junsoo Ha, and Gunhee Kim. "Harmonizing maximum likelihood with GANs for multimodal conditional generation." arXiv preprint arXiv:1902.09225 (2019).
>
> [c] - Arvanitidis, Georgios, Lars Kai Hansen, and Søren Hauberg. "Latent space oddity: on the curvature of deep generative models." arXiv preprint arXiv:1710.11379 (2017)
>
> [d] - https://en.wikipedia.org/wiki/Gnomonic_projection
>
> [e] - https://en.wikipedia.org/wiki/Equirectangular_projection
>
> [f] - Ramasinghe, Sameera, et al. "Conditional Generative Modeling via Learning the Latent Space." arXiv preprint arXiv:2010.03132 (2020).
>
> [g] - https://soochanlee.com/publications/mr-gan.html
>
> [h] - Geirhos, Robert, et al. "Shortcut learning in deep neural networks." Nature Machine Intelligence 2.11 (2020): 665-673.

---

> > ### Comment · Reviewer_q3Au · 2021-09-12
> > **Response to the review**
> >
> > Thanks for your detailed response.

---

### Decision · Program_Chairs · 2021-09-27

**Decision:**

Accept (Poster)

**Comment:**

The reviewers were in agreement that the method is of broad interest to the community, and is sufficiently general to improve many types of GANs. In the initial reviews, the author's noted that the GAN used was not sufficiently recent. The authors addressed this and other concerns more than satisfactorily, and all reviewers agree that this paper should be accepted. I agree. Nice work!